# Prenatal methadone exposure disrupts behavioral development and alters motor neuron intrinsic properties and local circuitry

Gregory G Grecco[1,2], Briana E Mork[1,3], Jui-Yen Huang[4,5], Corinne E Metzger[6], David L Haggerty[1], Kaitlin C Reeves[1], Yong Gao[1], Hunter Hoffman[1], Simon N Katner[7], Andrea R Masters[8], Cameron W Morris[1,9], Erin A Newell[7], Eric A Engleman[1], Anthony J Baucum[1,9,10], Jiuen Kim[1,10], Bryan K Yamamoto[1,10], Matthew R Allen[6,11], Yu-Chien Wu[10,12], Hui-Chen Lu[1,4], Patrick L Sheets[1,10], Brady K Atwood[1,10]*

[1]Department of Pharmacology and Toxicology, Indiana University School of Medicine, Indianapolis, United States; [2]Indiana University School of Medicine, Medical Scientist Training Program, Indianapolis, United States; [3]Program in Medical Neuroscience, Stark Neurosciences Research Institute, Indiana University School of Medicine, Indianapolis, United States; [4]Department of Psychological and Brain Sciences, Indiana University, Bloomington, United States; [5]The Linda and Jack Gill Center for Biomolecular Sciences, Department of Psychological and Brain Science, Program in Neuroscience, Indiana University, Bloomington, United States; [6]Department of Anatomy, Cell Biology & Physiology, Indiana University School of Medicine, Indianapolis, United States; [7]Deparment of Psychiatry, Indiana University School of Medicine, Indianapolis, United States; [8]Clinical Pharmacology Analytical Core-Indiana University Simon Cancer Center, Indiana University School of Medicine, Indianapolis, United States; [9]Department of Biology, Indiana University-Purdue University, Indianapolis, United States; [10]Stark Neurosciences Research Institute, Indiana University School of Medicine, Indianapolis, United States; [11]Indiana Center for Musculoskeletal Health, Indiana University School of Medicine, Indianapolis, United States; [12]Department of Radiology and Imaging Sciences, Indiana University School of Medicine, Indianapolis, United States

*For correspondence:
bkatwood@iu.edu

Competing interests: The authors declare that no competing interests exist.

**Abstract** Despite the rising prevalence of methadone treatment in pregnant women with opioid use disorder, the effects of methadone on neurobehavioral development remain unclear. We developed a translational mouse model of prenatal methadone exposure (PME) that resembles the typical pattern of opioid use by pregnant women who first use oxycodone then switch to methadone maintenance pharmacotherapy, and subsequently become pregnant while maintained on methadone. We investigated the effects of PME on physical development, sensorimotor behavior, and motor neuron properties using a multidisciplinary approach of physical, biochemical, and behavioral assessments along with brain slice electrophysiology and in vivo magnetic resonance imaging. Methadone accumulated in the placenta and fetal brain, but methadone levels in offspring dropped rapidly at birth which was associated with symptoms and behaviors consistent with neonatal opioid withdrawal. PME produced substantial impairments in offspring physical growth, activity in an open field, and sensorimotor milestone acquisition. Furthermore, these behavioral alterations were associated with reduced neuronal density in the motor cortex and a

disruption in motor neuron intrinsic properties and local circuit connectivity. The present study adds to the limited body of work examining PME by providing a comprehensive, translationally relevant characterization of how PME disrupts offspring physical and neurobehavioral development.

## Introduction

Pregnant women and their developing fetuses represent a vulnerable population severely impacted by the opioid crisis. From 1999 to 2014, the annual prevalence of opioid use disorder (OUD) in pregnant women at delivery increased 333% (*Haight et al., 2018*). Mirroring this rise, neonatal opioid withdrawal syndrome (NOWS) increased from 1.2 to 5.8 per 1000 hospital births from 2000 to 2012 with some geographical regions near 33 cases per 1000 (*Ko et al., 2016*; *Patrick et al., 2015*). Opioid maintenance therapies, such as methadone and buprenorphine, continue to represent the first-line treatments for pregnant women with OUD because these therapies benefit overall maternal-fetal outcomes at parturition (*ACOG, 2017*). However, it is unknown exactly how prenatal methadone exposure (PME) impacts the brain and behavioral development as these infants mature.

Prenatal opioid exposure is associated with smaller head circumferences (*Towers et al., 2019*), lower brain volumes (*Hartwell et al., 2020*; *Sirnes et al., 2017*), and deficits in white matter microstructure (*Monnelly et al., 2018*), although these studies are limited by small sample sizes making it difficult to control for confounding variables. A recent, large study using data from the Adolescent Brain Cognitive Development study reported reduced motor cortical volumes and surface area in children with prenatal opioid exposure (*Hartwell et al., 2020*). These differences remained significant when controlling for multiple additional factors (e.g. socioeconomic factors and prenatal alcohol/tobacco), indicating opioid exposure in utero may specifically disrupt motor cortex development and potentially impact motor behavior (*Hartwell et al., 2020*). Prior studies revealed poorer motor performance in children exposed to opioids prenatally suggesting that prenatal opioid exposure produces long-lasting changes in both motor behavior and in brain regions associated with motor functioning (*Lee et al., 2020*; *Yeoh et al., 2019*). However, clinical studies are often complicated by significant environmental variations which are difficult to control such as the extent of prenatal care or additional prenatal substance exposures which also impact neurodevelopmental outcomes of children (*Larson et al., 2019*). Therefore, a translationally relevant animal model of prenatal opioid exposure is desperately needed to examine the neurological and behavioral consequences of opioid exposure in the absence of confounding variables.

Animal models of prenatal opioid exposure have demonstrated deficits in several sensorimotor milestones (*Kunko et al., 1996*; *Robinson et al., 2020*; *Wallin et al., 2019*), but the mechanisms underlying disrupted sensorimotor development are unclear. Preclinical studies suggest that prenatal opioid exposure may prevent normal neuronal development (*Lu et al., 2012*; *Ricalde and Hammer, 1990*) and disturb myelination (*Jantzie et al., 2020*), which could underlie the aberrant neurobehavioral development. Unfortunately, the translational value of these rodent studies is limited by models that do not adequately model the majority of human prenatal opioid exposure cases. For instance, a large proportion of studies initiate opioid delivery around mid-gestation or later and often utilize morphine, even though methadone and buprenorphine account for a growing majority of prenatal opioid exposure cases (*Byrnes and Vassoler, 2018*; *Duffy et al., 2018*). Initiating opioid exposure at later stages of pregnancy overlooks the effect of opioids on earlier embryonic developmental processes. These weaknesses underlie the call for improved animal models of prenatal opioid exposure, which encompass all stages of prenatal brain development (*Larson et al., 2019*).

We developed a more translational mouse model that resembles the typical pattern of opioid use in a pregnant woman who is first dependent on oxycodone, then begins methadone maintenance treatment, and subsequently becomes pregnant while maintained on methadone. We found that PME reduces physical growth in offspring which persists into adolescence and disrupts the development of locomotor activity and ultrasonic vocalization (USV) during the preweaning period. Furthermore, PME delays the development of specific sensorimotor milestones. These impairments were concurrently associated with alterations in layer 5 motor cortical neuron intrinsic properties and local connectivity.

**eLife digest** The far-reaching opioid crisis extends to babies born to mothers who take prescription or illicit opioids during pregnancy. Opioids such as oxycodone and methadone can freely cross the placenta from mother to baby. With the rising misuse of and addiction to opioids, the number of babies born physically dependent on opioids has risen sharply over the last decade. Although these infants are only passively exposed to opioids in the womb, they can still experience withdrawal symptoms at birth. This withdrawal is characterized by irritability, excessive crying, body shakes, problems with feeding, fevers and diarrhea.

While considerable attention has been given to treating opioid withdrawal in newborn babies, little is known about how these children develop in their first years of life. This is, in part, because it is difficult for researchers to separate drug-related effects from other factors in a child's home environment that can also disrupt their development. In addition, the biological mechanisms underpinning opioid-related impairments to infant development also remain unclear.

Animal models have been used to study the effects of opioid exposure during pregnancy (termed prenatal exposure) on infants. These models, however, could be improved to better replicate the typical pattern of opioid use among pregnant women.

Recognizing this gap, Grecco et al. have developed a mouse model of prenatal methadone exposure where female mice that were previously dependent on oxycodone were treated with methadone throughout their pregnancy. Methadone is an opioid drug commonly prescribed for treating opioid use disorder in pregnant women and was found to accumulate at high levels in the fetal brain of mice, which fell quickly after birth. The offspring also experienced withdrawal symptoms. Grecco et al. then examined the physical, behavioral and brain development of mice born to opioid-treated mothers. These included assessments of the animals' motor skills, sensory reflexes and behavior in their first four weeks of life. Additional experiments tested the properties of nerve cells in the brain to examine cell-level changes.

The assessments showed that methadone exposure in the womb impaired the physical growth of offspring and this persisted into 'adolescence'. Prenatal methadone exposure also delayed progress towards key developmental milestones and led to hyperactivity in three-week-old mice. Moreover, Grecco et al. found that these mice had reduced neuron density and cell-to-cell connectivity in the part of the brain which controls movement.

These findings shed light on the potential consequences of prenatal methadone exposure on physical, behavioral and brain development in infants. This model could also be used to study new potential treatments or intervention strategies for offspring exposed to opioids during pregnancy.

## Results

### Impact of PME on pregnancy and litter characteristics

An overview of model development and workflow representing the total animals used for these studies is found in *Figure 1*. *Table 1* summarizes pregnancy and litter characteristics. Opioid treatment did not reduce pregnancy rates (chi square test: $\chi^2_{(1, \ n=95)}$=0.904, p=0.34) or instances of obstructed labor (chi square test: $\chi^2_{(1, \ n=70)}$=2.14, p=0.14). Litter sizes at birth (unpaired t test: $t_{61}$ = 1.36, p=0.18) and total neonatal deaths (chi square test: $\chi^2_{(1, \ n=406)}$=0.201, p=0.65) were not affected by opioid treatment. There was a trend for fewer PME males surviving to P7 (chi square test: $\chi^2_{(1, \ n=336)}$=2.71, p=0.10). However, the proportion of sexes at birth between treatment groups was not significantly different (chi square test: $\chi^2_{(1, \ n=284)}$=0.010, p=0.92) suggesting the reduction of male offspring at P7 may be attributed to the greater postnatal deaths in males relative to females in PME litters (chi square test: $\chi^2_{(1, \ n=37)}$=4.79, p=0.03).

### Methadone and EDDP concentrations in dams and offspring

Methadone and 2-ethylidene-1,5-dimethyl-3,3-diphenylpyrrolidine (EDDP, main metabolite of methadone) concentrations in the placenta, plasma, and brain for dams and offspring on G18, P1, and P7 are presented in *Supplementary file 1* and *Figure 2a*. Maternal brain and plasma levels were highest at G18 (248.9 ± 54.1 ng/g and 63.5 ± 11.3 ng/mL, respectively) and lower slightly after giving

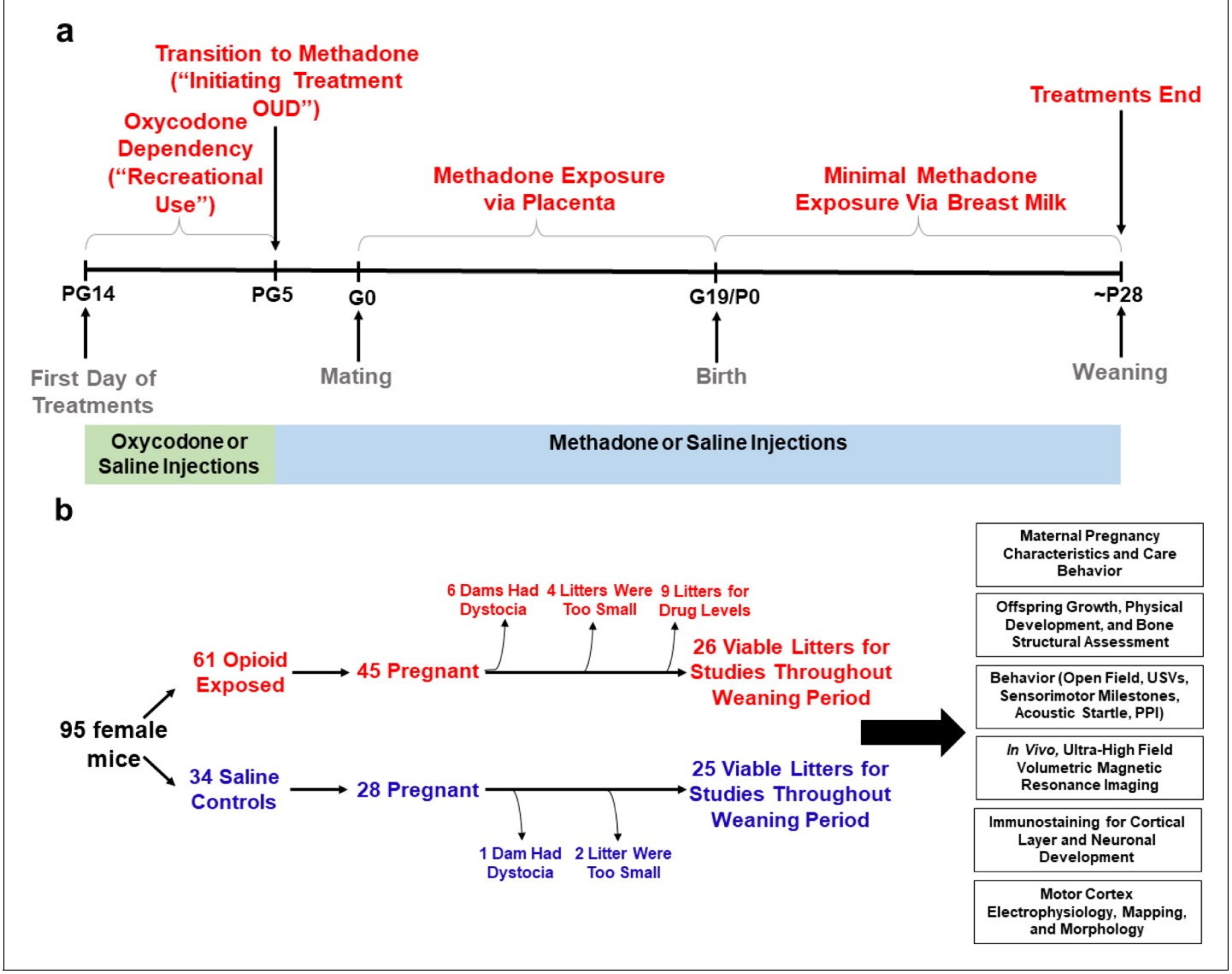

**Figure 1.** Study overview. (a) Timeline briefly describing the generation of mice with prenatal methadone exposure (PME). Approximately 15 days prior to mating (pregestational day 14), female mice begin treatment with oxycodone to model recreational opioid use. Following nine days of oxycodone injections, mice began receiving methadone injections to simulate treatment of opioid use disorder (OUD). Following 5 days of methadone administration, the females were mated (gestational day 0) and treatment continued throughout gestation passively exposing the developing embryo and fetus to methadone. Following birth, dam methadone treatment continued which provided minimal, but measurable methadone exposure to offspring (see *Figure 2*. and *Supplementary file 1*). Offspring were weaned at approximately postnatal day 28. All doses were given subcutaneously twice daily. Control animals underwent an identical timeline with the exception of receiving saline injections. (b) Flow chart reviewing studies completed in the present study using offspring generated from the timeline in (a). Female mice were randomly assigned to either opioid (oxycodone → methadone) or saline treatment. Of the 45 opioid treated mice which became pregnant, six were removed from the study due to obstructed labor (dystocia), four litters were too small (<3 pups), and nine were chosen at random to assess methadone tissue and plasma concentrations. Of the 28 pregnant control mice, one experienced dystocia and two litters were too small. The offspring from the remaining litters (25–26 per exposure) were pseudo-randomly allocated to the behavioral and biological studies indicated in the black boxes (≥4 litters per exposure with equal representation of both sexes were used for each experiment). See *Table 1* for detailed description of litter characteristics. *PG*, pregestation; *G*, gestation; *P*, postnatal, *OUD*, opioid use disorder; *USVs*, ultrasonic vocalizations; *PPI*, prepulse inhibition.

birth, remaining relatively stable at P1 and P7. Methadone accumulated in the brain relative to the plasma for dams at all timepoints (*Figure 2a*). The placenta significantly retained methadone (3862.1 ± 258.4 ng/g) and EDDP (1124.0 ± 84.4 ng/g; *Figure 2a,b*). The fetal brain methadone concentration was 2100.8 ± 237.6 ng/g which was substantially higher than the maternal brain on G18

**Table 1.** Pregnancy and litter characteristics.

All data were analyzed using Chi-Square tests except litter size which was analyzed using an unpaired, independent t test (value represents mean ± SEM). Sex was not determined in earlier cohorts of animals until nipple marks began to appear (~P7). However, given the potential sex differences in survival, for later cohorts we began determining sex at P0 and P7 which led to a lower total sample size of litters at the P0 date of sex determination. Postnatal deaths occur within approximately 72 hr after birth. As not all litters survived to ~P7 when sex was determined in offspring, the total sample size of litters from which postnatal deaths were examined is larger than the total sample size of litters for P7 sexes. *PME, Prenatal methadone exposure; PSE, Prenatal saline exposure, P0, P7, Postnatal day 0,7.*

| | PME | PSE | p-value |
|---|---|---|---|
| Total pregnancies (%) | 45/61 mated (73.8%) | 28/34 mated (82.4%) | p=0.34 |
| Obstructed labor (%) | 6/42 pregnancies (14.3%) | 1/27 pregnancies (3.6%) | p=0.14 |
| Litter size | 7.50 ± 0.34 (n = 36 litters) | 6.85 ± 0.31 (n = 27 litters) | p=0.18 |
| P0 Male:Female Ratio (% Male) | 96M:66F from 21 litters (59.3%) | 73M:49F from 18 litters (59.8%) | p=0.58 |
| P7 Male:Female Ratio (% Male) | 90M:90F from 26 litters (50.0%) | 92M:64F from 25 litters (59.0%) | p=0.10 |
| Postnatal deaths (%) | 50/248 from 33 litters (20.2%) | 29/158 from 27 litters (18.4%) | p=0.65 |
| Male:Female Postnatal Death Ratio (% Male Deaths) | 18M:5F from 21 litters (78.3%) | 6M:8F from 18 litters (42.9%) | **p=0.03** |

The online version of this article includes the following source data for Table 1:

Source data 1. Numerical data to support graphs in *Table 1* describing pregnancy and litter characteristics.

(*Figure 2a*). Offspring brain levels dropped precipitously to 7.9 ± 0.6 ng/g and 3.1 ± 0.3 ng/g on P1 and P7, respectively. Although we were not able to collect a sufficient plasma volume on G18, minimal methadone plasma levels in offspring were quantified postnatally. Placental levels were predictive of fetal brain methadone levels at G18 ($R^2$ = 0.39, p<0. 0078; *Figure 2c*). Offspring plasma methadone was predictive of offspring brain methadone at P1 ($R^2$ = 0.45, p<0.0084; *Figure 2d*), but not on P7 ($R^2$ = 0.008, p=0.74; *Figure 2e*).

## Maternal characteristics in opioid-treated dams

The opioid treatment strategy did not significantly impact maternal weights during the course of the study (rmANOVA: Treatment, $F_{(1,42)}$=0.00117, p=0.97; Time, $F_{(2.92,122.4)}$ = 266.1, p<0.0001; Interaction, $F_{(22,924)}$=0.510, p=0.97; *Figure 3a*). Measures of food consumption revealed a significant effect of opioid treatment (rmANOVA: Treatment, $F_{(1,33)}$=5.09, p=0.03; Time, $F_{(7,231)}$=413, p<0.0001; Interaction, $F_{(7,231)}$=2.48, p=0.02; *Figure 3b*) with an increased food consumption during the postnatal period in opioid-treated dams reaching significance on postnatal week 2 (Sidak's post-hoc test, p=0.004). Maternal care-related behavior on P3 was assessed to examine the potential impact of methadone on offspring care. There were no apparent differences in nest quality (Mann-Whitney test: U = 53, p=0.15; *Figure 3c*) or average latency to retrieve pups removed from the nest (Mann-Whitney test: U = 77, p=0.97; *Figure 3d*). No placentas remained in the cages 24 hr after birth in either group indicating normal placentophagy. Oxycodone treatment prior to gestation (PG14-PG6; see *Figure 1a*) induced opioid dependency in dams as demonstrated by the significantly increased withdrawal behaviors following naloxone treatment (*Figure 3e*). Offspring from mothers that were assessed for oxycodone dependency were not used in subsequent analyses to avoid any impact that pregestational opioid withdrawal could have on offspring outcomes. Methadone maintained opioid dependency as evidenced by the significantly increased naloxone-precipitated withdrawal behaviors observed in opioid-treated females at post-weaning (ANOVA: Treatment, $F_{(1,26)}$=38.1, p<0.0001; Stage, $F_{(1,26)}$=0.361, p=0.55; Interaction, $F_{(1,26)}$=0.337, p=0.56; *Figure 3e*).

## Physical development of offspring

The clinical evidence for the effect of prenatal opioid exposure on fetal growth has revealed mixed findings (*Yazdy et al., 2015*); therefore, we first examined the impact of PME on offspring physical development. As there were no significant effects of sex, the data on weight and lengths during the pre-weaning period were pooled. PME significantly reduced weights and reduced weight gain during the pre-weaning period (rmANOVA: Exposure, $F_{(1,111)}$=3.96, p=0.049; Time, $F_{(1.344,149.2)}$ = 3177, p<0.0001; Interaction, $F_{(6,666)}$=2.89, p=0.0087; *Figure 4a*). The reduced body weight in PME

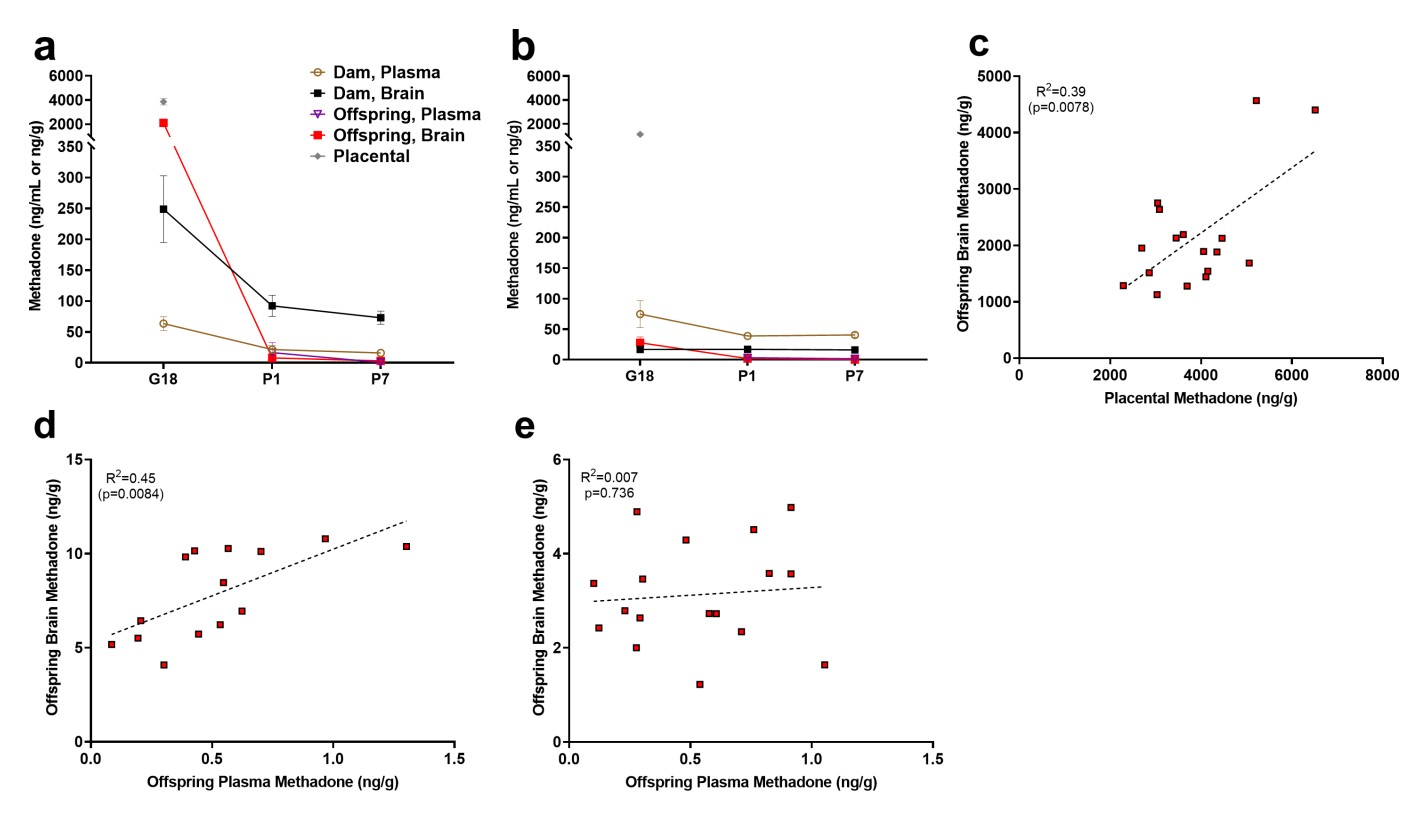

**Figure 2.** Relationship between placental, plasma, and brain methadone and metabolite levels in dams and offspring during gestation and the postnatal period. (**a**) Methadone and (**b**) EDDP (2-ethylidene-1,5-dimethyl-3,3-diphenylpyrrolidine, main metabolite of methadone) concentrations in the plasma, brain, and placenta of dams and offspring on G18 (approximately 1 day before birth), P1 (approximately 1 day after birth), and P7. Methadone highly accumulated in the fetal compartment relative to dam concentrations, but methadone concentrations dropped precipitously following birth. EDDP accumulated in the placenta. Data points indicate mean ± SEM. (**c**) Placental methadone concentrations predicted fetal brain concentrations on G18 ($R^2$ = 0.39, p=0.0078). (**d,e**) Offspring plasma methadone concentrations predicted offspring brain methadone on P1 ($R^2$ = 0.45, p=0.0084), but not on P7 ($R^2$ = 0.008, p=0.736). All tissue and blood samples were collected 2.5 hr following the morning administration of methadone. (n = 3 dams + their respective litters per timepoint; n = 17–20 offspring samples at G18, n = 15 offspring at P1, n = 17–18 offspring at P7). Data are collapsed across offspring sex. The limit of quantification for methadone and EDDP detection was 0.1 ng/mL and 0.05 ng/mL in the plasma, respectively, and 0.08 ng/sample and 0.04 ng/sample of placenta and brain for both methadone and EDDP.

The online version of this article includes the following source data for figure 2:

**Source data 1.** Numerical data to support graphs in *Figure 2* describing placental, plasma, and brain methadone and metabolite levels in dams and offspring.

offspring persisted into adolescence as weights remained consistently lower across sexes at P35 and P49 (rmANOVA: Exposure, $F_{(1,50)}$=7.52, p=0.0085; Sex, $F_{(1,50)}$=131, p<0.0001, Time, $F_{(1,50)}$=673, p<0.0001; Time x Sex, $F_{(1,50)}$=62.3, p<0.0001; Time x Exposure, $F_{(1,50)}$=1.60, p=0.21; Exposure x Sex, $F_{(1,50)}$=0.00956, p=0.92; Time x Exposure x Sex, $F_{(1,50)}$=0.20, p=0.66; *Figure 4a*). Similarly, body length was reduced by PME (rmANOVA: Exposure, $F_{(1,111)}$=12.24, p=0.0007; Time, $F_{(2.676,297.0)}$ = 4719, p<0.0001; Interaction, $F_{(6,666)}$=0.987, p=0.433; *Figure 4b*).

As chronic opioid use is associated with bone pathology (*Mattia et al., 2012*), we collected femurs from P7 and P35 offspring to assess bone structure and density. Although femur length was not impacted by PME (unpaired t test: $t_{23}$ = 1.126, p=0.27; *Figure 4—figure supplement 1a*), whole bone volume tended to be lower in P7 PME offspring (unpaired t test: $t_{23}$ = 1.89, p=0.072; *Figure 4—figure supplement 1b*). By P35, structural bone measures were similar between exposure groups including distal femur metaphysis bone volume (ANOVA: Exposure, $F_{(1,16)}$=1.94, p=0.18; Sex, $F_{(1,16)}$=13.4, p=0.0021; Interaction, $F_{(1,16)}$=2.17, p=0.16; *Figure 4—figure supplement 1c*), trabecular bone volume (ANOVA: Exposure, $F_{(1,16)}$=1.30, p=0.27; Sex, $F_{(1,16)}$=25.7, p=0.0001; Interaction, $F_{(1,16)}$=0.126, p=0.73; *Figure 4—figure supplement 1d*), cortical bone area (ANOVA:

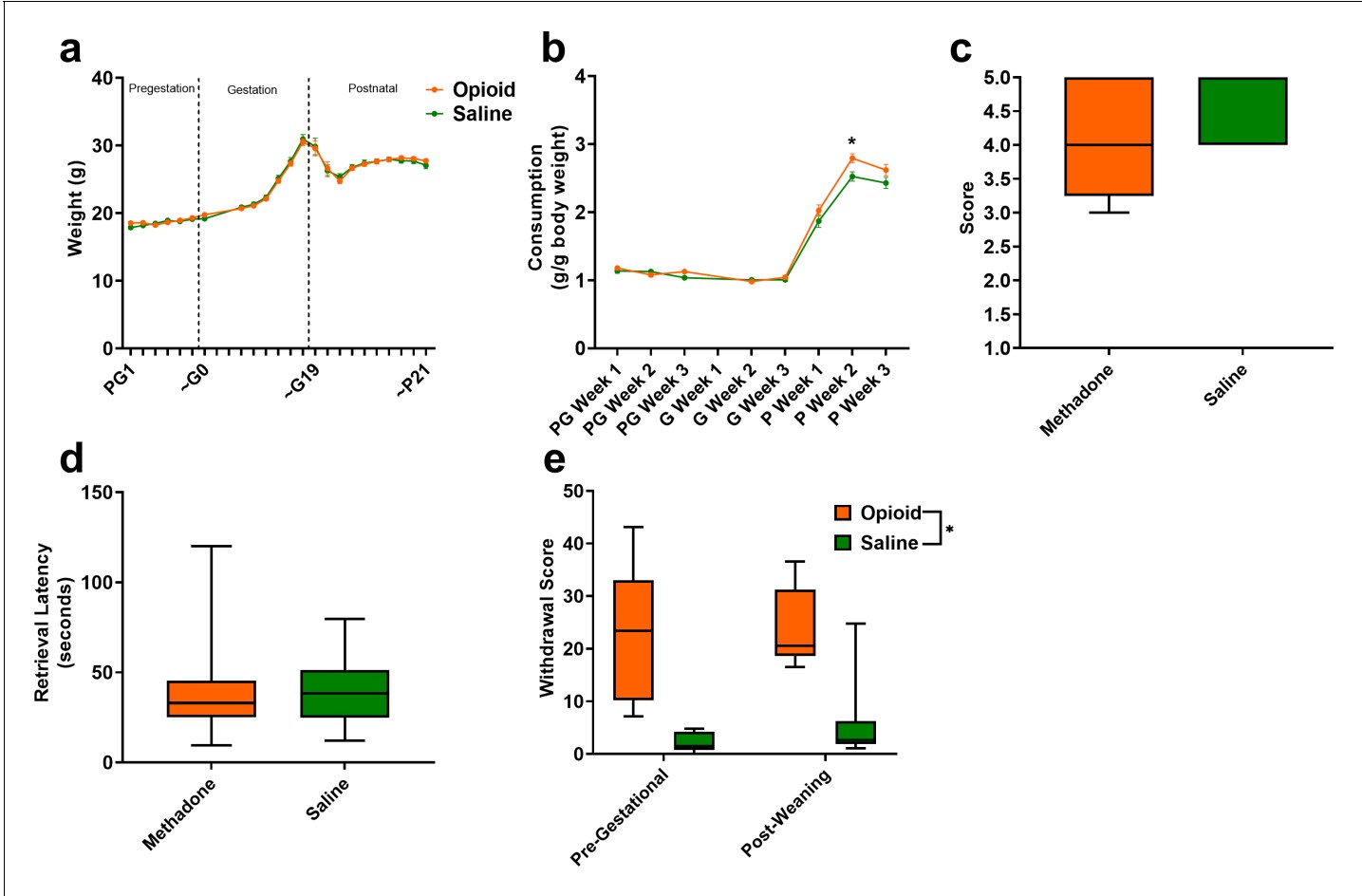

**Figure 3.** Opioid treatment-induced dependency and reduced gestational weight gain but did not alter maternal care. (a) Measures of maternal weights over the course of the study revealed no significant effects of opioid treatment (n = 23 opioid, 21 saline mice); however, (b) Maternal food consumption was not significantly affected during pregestation and gestation but was significantly increased in opioid-treated dams following birth (rmANOVA: Interaction, p=0.02; Sidak's post hoc: P week 2, p=0.004, n = 19 opioid, 16 saline mice). (c,d) Opioid-treated dams demonstrated no differences in (c) nest quality or (d) latency to retrieve pups removed from the nest on P3 suggesting maternal care behavior was grossly intact (n = 12 opioid, 13 saline mice). (e) The nine days of pregestational oxycodone treatment induced maternal opioid dependency and the oxycodone treatment plus the transition to methadone throughout gestation and the pre-weaning period was sufficient to maintain opioid dependence as determined by measures of naloxone-precipitated withdrawal behaviors (ANOVA: Treatment, p<0.0001, n = 7–8 opioid, 7–8 saline mice). *p<0.05. (a,b) Data points indicate mean ± SEM. (c–e) box plots indicate 25th to 75th percentiles with whisker characterizing the minimum and maximum value.

The online version of this article includes the following source data for figure 3:

**Source data 1.** Numerical data to support graphs in *Figure 3* describing maternal characteristics and care behavior.

Exposure, $F_{(1,16)}$=0.000, p=0.99; Sex, $F_{(1,16)}$=13.4, p=0.0021; Interaction, $F_{(1,16)}$=2.04, p=0.17; *Figure 4—figure supplement 1e*), and cortical thickness (ANOVA: Exposure, $F_{(1,16)}$=0.0435, p=0.84; Sex, $F_{(1,16)}$=19.0, p=0.0005; Interaction, $F_{(1,16)}$=0.790, p=0.39; *Figure 4—figure supplement 1f*).

A recent report suggested opioid-exposed infants have a higher prevalence of orofacial clefting which may reflect an effect of prenatal drug exposure on craniofacial development (*Mullens et al., 2019*). However, we did not observe any effect on the day both eyes opened (unpaired t test: $t_{49}$ = 1.67, p=0.10), both ears moved to their final erect position and external auditory canals were patent (unpaired t test: $t_{49}$ = 0.910, p=0.37), or when bottom incisor teeth erupted (unpaired t test: $t_{49}$ = 1.05, p=0.30; *Figure 4—figure supplement 2*) indicating measures of mouse postnatal craniofacial development progress normally in our PME model.

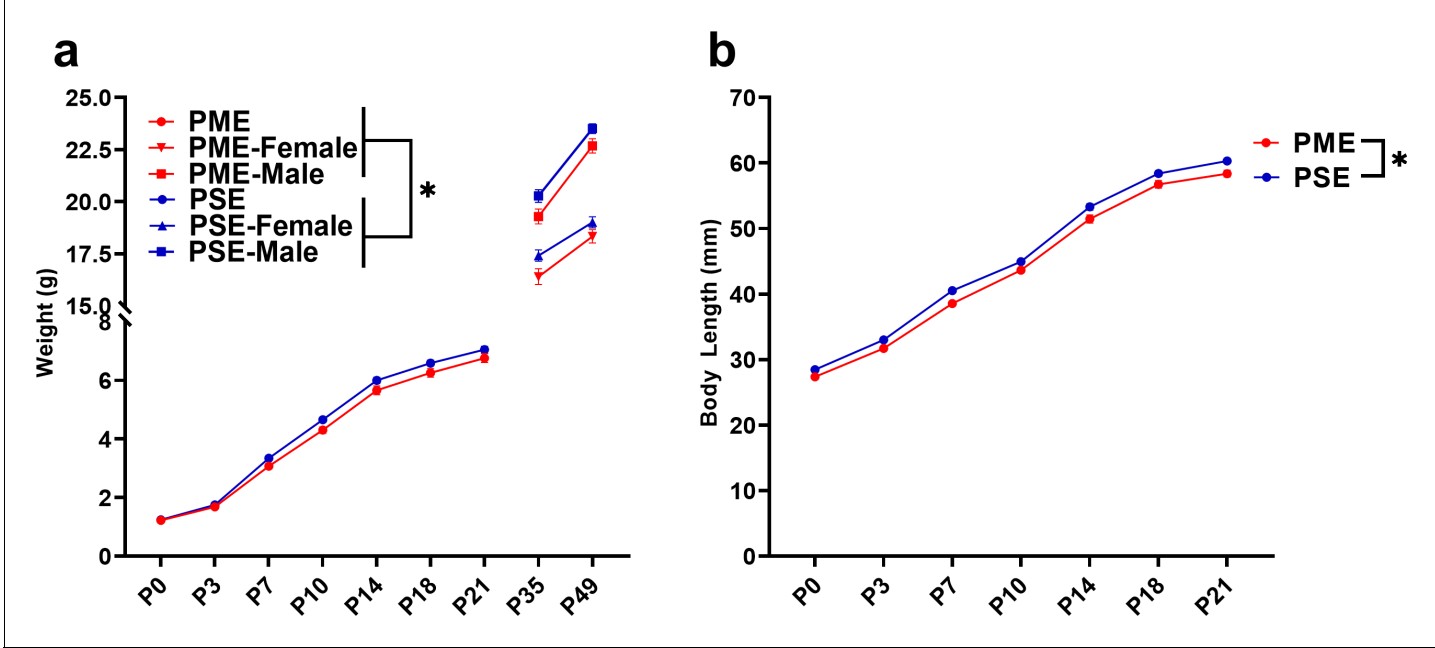

**Figure 4.** PME impaired offspring physical development. (a) PME reduced offspring weight during the preweaning period (rmANOVA: Exposure, p=0.049; Interaction, p=0.0087, n = 57 PME (25M:32F), 56 PSE (29M:27F)) and this effect on weight persisted into adolescence in both sexes (rmANOVA: Exposure, p=0.0085, n = 30 PME (16M:14F), 24 PSE (15M:9F)). (b) PME reduced offspring body length during the preweaning period (rmANOVA: Exposure, p=0.0007, n = 57 PME (25M:32F), 56 PSE (29M:27F)). *main effect of exposure p<0.05. From P0-P21, Data were collapsed on sex as initial analyses revealed no main-effects or interaction with sex. Data points indicate mean ± SEM.

The online version of this article includes the following source data and figure supplement(s) for figure 4:

**Source data 1.** Numerical data to support graphs in *Figure 4* describing physical development in offspring.

**Figure supplement 1.** Early life bone density may be reduced in prenatal methadone-exposed offspring but recovered by adolescence.

**Figure supplement 1—source data 1.** Numerical data to support graphs in *Figure 4—figure supplement 1* describing bone development in offspring.

**Figure supplement 2.** Postnatal craniofacial milestones in offspring is not affected by PME.

**Figure supplement 2—source data 1.** Numerical data to support graphs in *Figure 4—figure supplement 2* describing craniofacial development in offspring.

## Behavioral development in offspring

As methadone levels rapidly dropped from G18 to P1 in offspring, we sought to determine if PME offspring demonstrated behaviors consistent with NOWS such as hyperthermia or myoclonic jerks (twitching or jerking of the limbs or whole-body) which are indicators of NOWS in humans (*Kocherlakota, 2014*). PME offspring displayed a relative hyperthermia at baseline that was maintained after two minutes of isolation (rmANOVA: Exposure, $F_{(1,48)}$=34.1, p<0.0001; Time, $F_{(1,48)}$=577, p<0.0001; Interaction, $F_{(1,48)}$=0.127, p=0.723; *Figure 5a*). A higher number of twitches/jerks were observed in P1 PME offspring (unpaired t test: $t_{49}$ = 2.04, p=0.047; *Figure 5b*) reminiscent of the myoclonic jerks which are common among human neonates experiencing NOWS from methadone (*Kocherlakota, 2014*). In association with a rapid drop in brain methadone levels from G18 to P1, the relative hyperthermia and increased twitches/jerks suggests PME offspring may experience opioid withdrawal at P1.

Preclinical studies have indicated locomotor activity may be different in prenatal opioid-exposed offspring (*Andersen et al., 2020*); however, no studies to date have examined the trajectory of locomotor development during early life in opioid-exposed offspring. Using an open field, we measured the development of motor activity by repeatedly testing animals on P1, P7, P14, and P21. Locomotor activity was significantly altered during the pre-weaning period in PME offspring (rmANOVA: Exposure, $F_{(1,49)}$=5.18, p=0.027; Time, $F_{(1.38,67.84)}$ = 299, p<0.0001; Interaction, $F_{(3,147)}$=10.5, p<0.0001) with PME offspring demonstrating significantly reduced activity at P1 but greater activity at P7 and P21 (Sidak's post hoc test: p=0.048, p=0.037, p=0.009, respectively; *Figure 5c*).

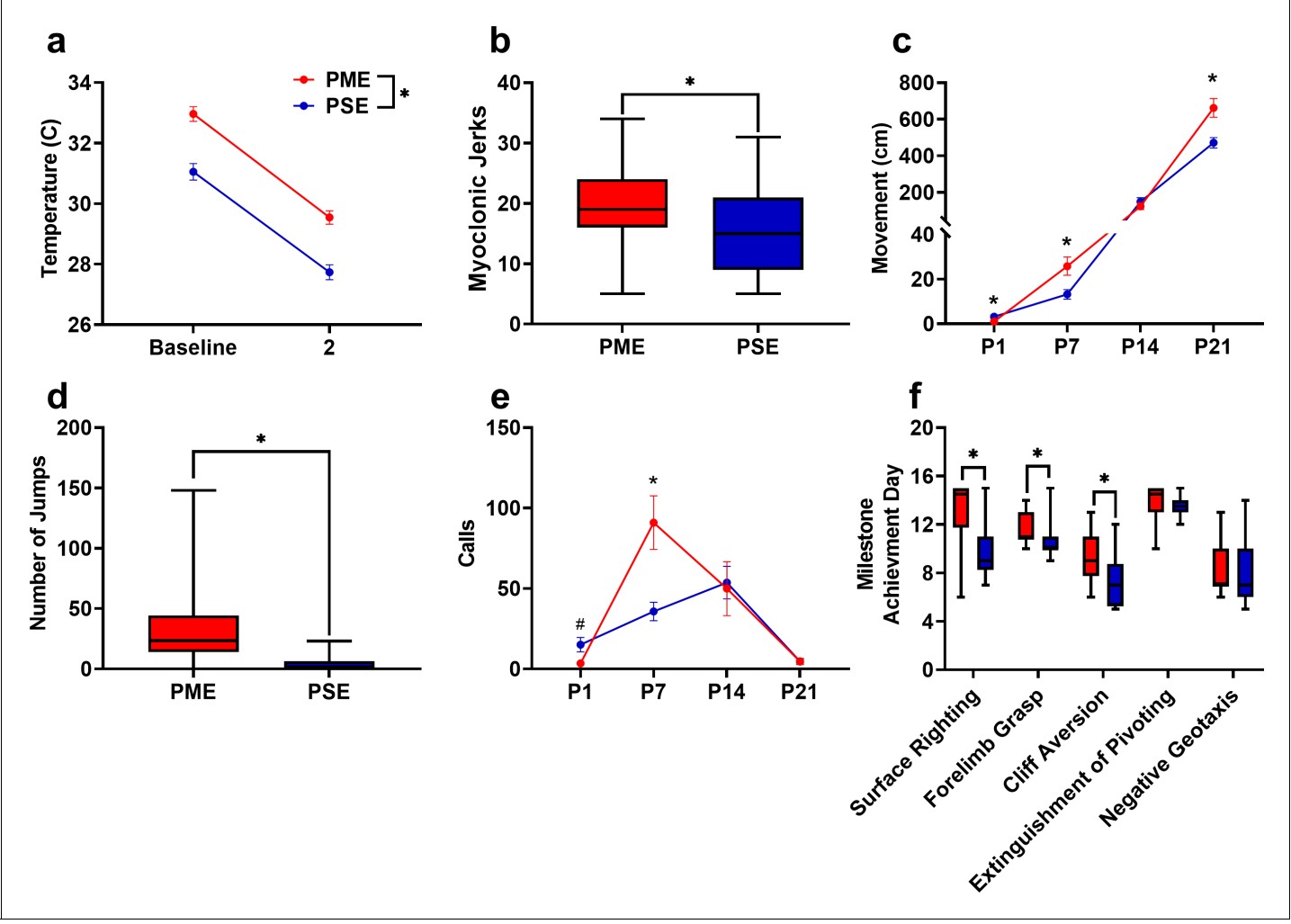

**Figure 5.** PME disrupted behavioral development in offspring. (a) Offspring showed significantly higher surface body temperature when removed from the nest (baseline) and continued to maintain a higher surface body temperature following two minutes of isolation at P1 (rmANOVA: Exposure, p<0.0001, n = 25 (7M:18F) PME, 25 PSE (15M:10F) mice). (b) PME offspring show a greater number of twitches/jerks at P1 similar to the myoclonic jerks reportedly observed in human neonates experiencing opioid withdrawal (unpaired t test: p=0.047, n = 26 (7M:19F) PME, 25 PSE (15M:10F) mice). (c) Offspring were repeatedly tested in a modified open field during the first three weeks of life to examine the development of locomotor activity and ultrasonic vocalization (USV) production. (c) PME altered the development of locomotor activity (rmANOVA: Exposure, p=0.027; Interaction, p<0.0001) with PME offspring showing reduced activity at P1 but significantly greater activity at P7 and P21 (Sidak's post hoc test: p=0.048, p=0.037, p=0.009, respectively, n = 26 (7M:19F) PME, 25 PSE (15M:10F) mice). (d) Subsequent manual scoring of videos on P21 revealed significantly greater number of vertical jumps during the 5-min session in PME mice further supporting a hyperactive phenotype at P21 (Mann-Whitney test: p<0.0001, n = 26 (7M:19F) PME, 25 PSE (15M:10F) mice). (e) PME offspring showed differences in the production of separation-induced USV calls (rmANOVA: Interaction, p=0.032) with less total calls emitted on P1 and significantly more calls emitted on P7 (Sidak's post hoc test: p=0.079, p=0.015, n = 26 (7M:19F) PME, 25 PSE (15M:10F) mice). (f). Acquisition of the surface righting (unpaired t test: p<0.0001), forelimb grasp, and cliff aversion (Mann-Whitney test: p=0.0029 and p=0.0026, respectively) were significantly delayed in PME offspring (n = 26 (7M:19F) PME, 24 PSE (14M:10F) mice). *p<0.05, # p=0.079. Data were collapsed on sex as initial analyses revealed no main-effects or interaction with sex with the exception of USV results (See *Figure 5—figure supplement 1* for results separated by sex). Data points indicate mean ± SEM. Box plots indicate 25–75th percentiles with whisker characterizing the minimum and maximum value.

The online version of this article includes the following source data and figure supplement(s) for figure 5:

**Source data 1.** Numerical data to support graphs in *Figure 5* describing behavioral development in offspring.

**Figure supplement 1.** Prenatal methadone exposed offspring displayed a greater propensity to fall over on postnatal day 1.

**Figure supplement 1—source data 1.** Numerical data to support graphs in *Figure 5—figure supplement 1* describing P1 activity in offspring.

**Figure supplement 2.** Open-field behaviors are altered in offspring with PME.

**Figure supplement 2—source data 1.** Numerical data to support graphs in *Figure 5—figure supplement 2* describing open-field behaviors in offspring.

*Figure 5 continued on next page*

*Figure 5 continued*

**Figure supplement 3.** Separation induced ultrasonic vocalizations separated by sex.
**Figure supplement 3—source data 1.** Numerical data to support graphs in *Figure 5—figure supplement 3* describing sex-specific USVs in offspring.
**Figure supplement 4.** In depth analysis of ultrasonic vocalization revealed differential patterns of call types and syntax at P7.
**Figure supplement 4—source data 1.** Numerical data to support graphs in *Figure 5—figure supplement 4* describing the in-depth analysis of P7 USVs in offspring.
**Figure supplement 5.** Acoustic startle response and prepulse inhibition was not affected by PME.
**Figure supplement 5—source data 1.** Numerical data to support graphs in *Figure 5—figure supplement 5* describing PPI and acoustic startle response in offspring.

Inspection of the P1 videos indicated the reduction in motor activity was likely a result of poor coordination as PME displayed a greater propensity to fall over earlier in the testing session. Although 88% of PME and 80% of PSE offspring fell over by the end of five-minute session on P1 (chi square test: $\chi2_{(1, n=51)}$=0.690, p=0.41; *Figure 5—figure supplement 1a*), 53.5% of PME compared to 28% of PSE animals had fallen over immediately upon placement into the arena (chi square test: $\chi^2_{(1, n=51)}$=0.30.52, p=0.061; *Figure 5—figure supplement 1b*). There was a nonsignificant reduction in the latency to fall over in PME offspring (Mann-Whitney test: U = 248.5, p=0.136; *Figure 5—figure supplement 1c*).

In addition to the hyperactivity on P21, PME offspring exhibited significantly more vertical jumps at P21 (Mann-Whitney test: U = 53, p<0.0001; *Figure 5d*) further indicating a hyperactive phenotype is present in juvenile PME offspring similar to what has been reported in opioid-exposed children (*Azuine et al., 2019*; *Sundelin Wahlsten and Sarman, 2013*). These jumps may represent escape attempts as some animals were able to jump and reach the top of the arena walls. When this occurred, the animal was outside of the frame of video and our tracking software was not able to record activity. Therefore, the total distance traveled at P21 in the PME animals may actually be greater than we are able to report. As increased anxiety-like behavior has also been reported in some prenatal opioid exposure models (*Andersen et al., 2020*; *Byrnes and Vassoler, 2018*), we further examined open field videos for thigmotaxis, grooming, and rearing behaviors. Thigmotaxis was affected by PME (rmANOVA: Exposure, $F_{(1,49)}$=0.17.33, p=0.0001; Time, $F_{(1.41,68.9)}$ = 236.3, p<0.0001; Interaction, $F_{(2,98)}$=12.01, p<0.0001), with PME animals spending significantly more time at P7 near the arena walls than PSE animals (Sidak's post hoc test: p=0.0004 *Figure 5—figure supplement 2a*) However, this likely reflects the relative hypoactivity of PSE animals which demonstrated very little activity on P7 and mostly remained in the arena center where offspring were placed at the beginning of each trial. Both grooming (Mann-Whitney test: U = 173.5, p=0.0037) and unsupported rearing (unpaired t test: $t_{49}$ = 2.48, p=0.017), but not supported rearing (unpaired t test: $t_{49}$ = 0.352, p=0.73) were reduced in PME mice which likely reflects an increase in time spent in active locomotion and jumping instead of grooming and rearing in PME mice as opposed to anxiolysis (*Figure 5—figure supplement 2b–d*).

When removed from the dam, pups emit separation-induced USVs which signal distress and encourage retrieving behavior from the mother. USVs typically peak around P8 and extinguish by the third week of life (*Ehret, 2005*). The production of USVs in the open field was also significantly altered by PME (rmANOVA: Exposure, $F_{(1,47)}$=0.380, p=0.54; Time, $F_{(3,141)}$=13.3, p<0.0001; Sex, $F_{(1,47)}$=4.24, p=0.045; Time x Exposure, $F_{(3,141)}$=3.01, p=0.032; Time x Sex, $F_{(3,141)}$=0.636, p=0.59; Exposure x Sex, $F_{(1,47)}$=1.57, p=0.22; Time x Exposure x Sex, $F_{(3,141)}$=0.947, p=0.42; *Figure 5—figure supplement 3* and collapsed on sex in *Figure 5e*). Female PME offspring vocalized significantly more than both male and female PSE offspring at P7 (Sidak's post hoc test; p=0.0045 and p=0.026, respectively) but this difference was not observed in male PME offspring relative to either sex of PSE offspring (Sidak's post hoc test; both p>0.999, *Figure 5—figure supplement 3*). When collapsed on sex, vocalizations are nonsignificantly reduced at P1 and significantly increased at P7 in PME offspring (Sidak's post hoc test: p=0.079 and p=0.015, respectively; *Figure 5e*). Although female PME offspring do not produce significantly more USVs than male PME offspring at P7 (Sidak's post hoc: p=0.60), the sex-collapsed effect described on P7 is likely driven by the high rate of USV production in female PME offspring (*Figure 5—figure supplement 3*). We next performed further analysis on P7 USVs as this day differed between exposure groups. Total activity correlated significantly with total calls on P7 across all animals (Pearson's r = 0.41, p=0.0028; *Figure 5—figure supplement 4a*)

suggesting that there may be a relationship between hyperactivity and hyper-vocalizations among PME offspring. USV call types were classified into categories based on frequencies, to cluster the types of USVs emitted on P7 (*Coffey et al., 2019*). The call classification analysis revealed the increase in USVs on P7 may be a result of a higher number of complex, flat, step down, and upward ramp calls emitted by PME offspring (unpaired t tests: $t_{49}$ = 2.54, p=0.014; $t_{49}$ = 2.98, p=0.0045; $t_{49}$ = 2.74, p=0.0086; $t_{49}$ = 2.09, p=0.041; *Figure 5—figure supplement 4b*). Markov chains of call type transitions ('syntax') was further analyzed on P7 which also revealed differences in the transition between call types between exposure groups (*Figure 5—figure supplement 4c*).

Beginning on P3 and through P14, offspring began a battery of daily developmental tests assessing the development of sensorimotor milestones. Offspring with PME demonstrated delayed acquisition of surface righting (unpaired t test: $t_{48}$ = 4.69, p<0.0001), forelimb grasp (Mann-Whitney test: U = 166, p=0.0029), and cliff aversion (Mann-Whitney test: U = 161, p=0.0026; *Figure 5f*). No differences in the acquisition of the extinguishment of pivoting (Mann-Whitney test: U = 238, p=0.14) or negative geotaxis behavior (Mann-Whitney test: U = 270.5, p=0.42) were observed. As basic locomotor activity is not impaired in PME offspring, these data indicate that PME offspring lack the ability to produce more complex, coordinated motor behaviors in response to multimodal sensory input.

A recent investigation of prenatal THC exposure revealed deficits in sensorimotor gating (*Frau et al., 2019*); therefore, as sensorimotor performance was impaired in PME offspring, we subsequently tested prepulse inhibition (PPI), a sensorimotor gating task. In a cohort of P28-P29 offspring, we found that PME did not alter the acoustic startle response (rmANOVA: Exposure, $F_{(1,36)}$=0.129, p=0.72; Intensity, $F_{(1.830,65.87)}$ = 185.5, p<0.0001; Interaction, $F_{(6,216)}$=0.147, p=0.99; *Figure 5—figure supplement 5a*). No exposure-related effects were found on the percent PPI suggesting sensorimotor gating is intact in this model (rmANOVA: Exposure, $F_{(1,36)}$=0.282, p=0.60; Prepulse Intensity, $F_{(1.860,66.97)}$ = 30.8, p<0.0001; Interaction, $F_{(2,72)}$=0.770, p=0.47; *Figure 5—figure supplement 5b*).

## Brain anatomical development in offspring

Numerous brain regions and neural mechanisms could contribute to the altered activity and sensorimotor development described in PME offspring; therefore, we began our investigation by probing structural differences in gray matter regions across the brain using in vivo, ultra-high-field volumetric MRI. Volumes of interest (VOIs; See *Figure 6—figure supplement 1* for segmentations) were used to determine if differences in various gray matter regions were present in offspring at P28-P30. No significant differences in VOIs across any of the brain regions examined were discovered suggesting gross gray matter structure is mostly unaffected by PME (unpaired t tests: see *Supplementary file 2* for full test statistics; *Figure 6*).

To determine whether PME could alter cortical development, we examined cortical laminations and neuron densities in distinctive cortical layers of P22-P24 offspring with triple immunostaining of NeuN (neuronal marker), Cux1 (a cortical layer 2/3–4 marker), and Draq5 (nuclei stain). We focused on the anterior cingulate cortex (ACC), primary somatosensory cortex (S1), and primary motor cortex (M1) for their involvement in sensorimotor behavior (*Hatsopoulos and Suminski, 2011*; *Paus, 2001*; *Umeda et al., 2019*; *Figure 7*, *Figure 7—figure supplement 1*, *Figure 7—figure supplement 2*). In M1, there was no difference in the distributions of Cux1$^+$ cells (ANOVA: Exposure, $F_{(1,34)}$=0.0943, p=0.76; Sex, $F_{(1,34)}$=1.68, p=0.20; Interaction, $F_{(1,34)}$=0.334, p=0.57; *Figure 7a–c*). Interestingly, we found a significant reduction in NeuN densities in cortical layers 2/3–4 (ANOVA: Exposure, $F_{(1,34)}$=4.40, p=0.044; Sex, $F_{(1,34)}$=7.56, p=0.0095; Interaction, $F_{(1,34)}$=5.014, p=0.032; *Figure 7a,b,d*) that was specific to PME females (Sidak's post hoc: p=0.0098). However, NeuN densities were not reduced in layer 5 (ANOVA: Exposure, $F_{(1,34)}$=0.673, p=0.42; Sex, $F_{(1,34)}$=0.733, p=0.40; Interaction, $F_{(1,34)}$=1.31, p=0.26; *Figure 7a,b,e*). In the ACC, there was a minor but significant Exposure x Bin interaction for both Cux1+ (rmANOVA: Exposure, $F_{(1,32)}$=1.03, p=0.32; Bin, $F_{(2.13,68.3)}$ = 394.6, p<0.0001; Interaction, $F_{(9,288)}$=2.12, p=0.028; *Figure 7—figure supplement 1a–c*) and NeuN+ cell densities (rmANOVA: Exposure, $F_{(1,32)}$=0.424, p=0.52; Bin, $F_{(4.88,156.0)}$ = 142.2, p<0.0001; Interaction, $F_{(9,288)}$=2.59, p=0.0070; *Figure 7—figure supplement 1a,b,d*). Although no posthoc tests quite reached the level of significance, there appeared to be an increase in NeuN+ cell densities in Bin 9 (corresponding to approximately to the layer 5/6 border; Sidak's post hoc: p=0.050; *Figure 7—figure supplement 1a,b,d*). In S1, the distribution patterns of Cux1 did not differ (rmANOVA:

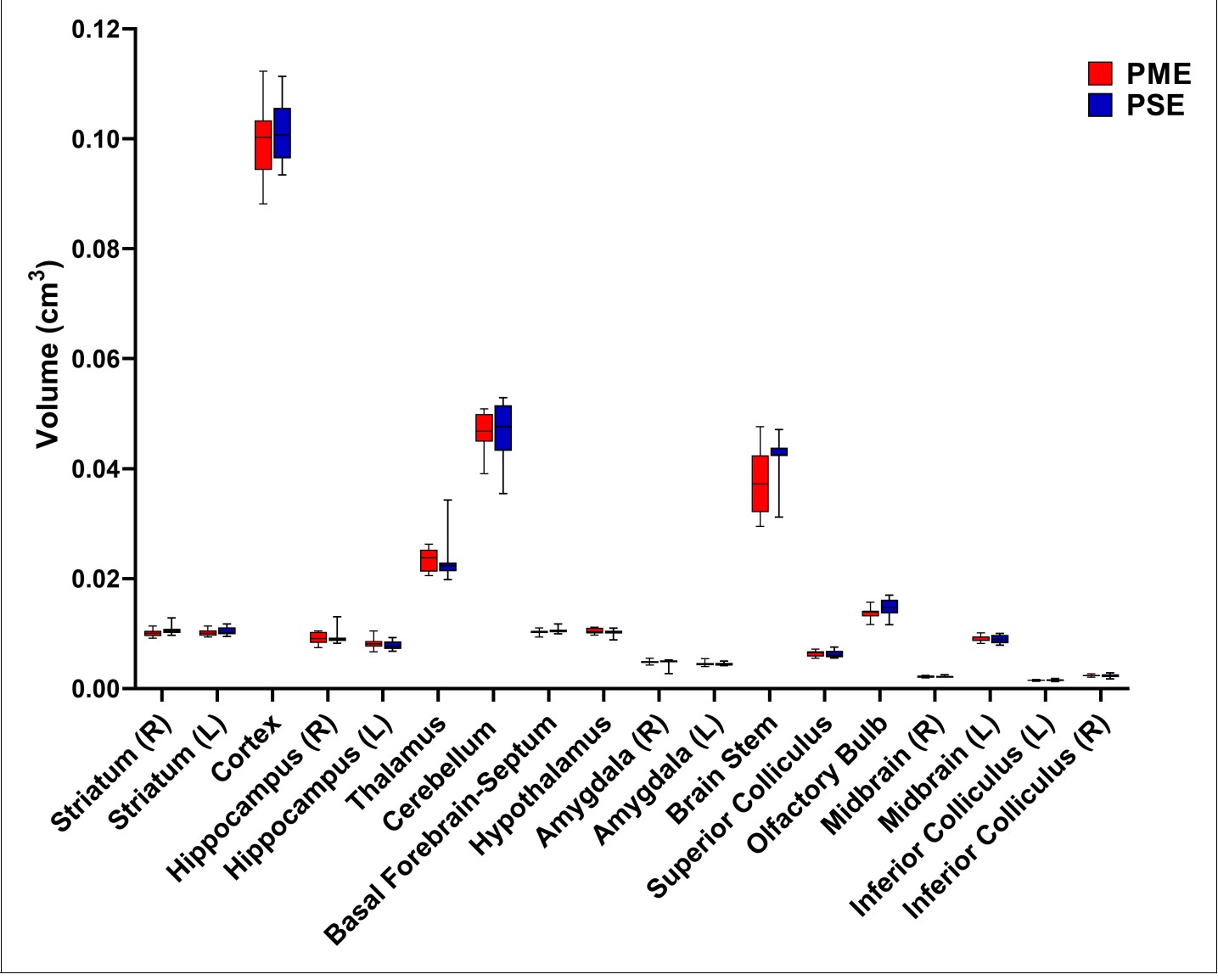

**Figure 6.** PME did not reduce volumes in brain regions of interest. Volumetric findings from ultra-high-field MRI of P28-P30 offspring indicate that gross brain structure is primarily unaffected for both cortical and subcortical regions in PME offspring (unpaired t tests, n = 11 (4M:7F) PME, 11 PSE (6M:5F) mice). Data were collapsed on sex due to underpowering. Box plots indicate 25–75th percentiles with whisker characterizing the minimum and maximum value.

The online version of this article includes the following source data and figure supplement(s) for figure 6:

**Source data 1.** Numerical data to support graphs in *Figure 6* and *Supplementary file 2* describing MRI gray matter volumes.

**Figure supplement 1.** Brain regions examined for volume differences in offspring.

Exposure, $F_{(1,310)}=0.498$, p=0.48; Bin, $F_{(9,310)}=346.5$, p<0.0001; Sex, $F_{(1,310)}=1.24$, p=0.27; Bin x Exposure, $F_{(9,310)}=0.860$, p=0.56; Bin x Sex, $F_{(9,310)}=1.54$, p=0.13; Exposure x Sex, $F_{(1,310)}=1.21$, p=0.27; Bin x Exposure x Sex, $F_{(9,310)}=1.72$, p=0.08; *Figure 7—figure supplement 2a–c*); however, there was a slight exposure related effect on NeuN+ cell densities as a there was a complex three way interaction between Exposure, Bin, and Sex (rmANOVA: Exposure, $F_{(1,310)}=3.30$, p=0.07; Bin, $F_{(9,310)}=63.6$, p<0.0001; Sex, $F_{(1,310)}=2.72$, p=0.10; Bin x Exposure, $F_{(9,310)}=0.811$, p=0.61; Bin x Sex, $F_{(9,310)}=0.840$, p=0.58; Exposure x Sex, $F_{(1,310)}=0.0007$, p=0.98; Bin x Exposure x Sex, $F_{(9,310)}=2.68$, p=0.0051; *Figure 7—figure supplement 2a,b,d*). Similar to the ACC, no posthoc tests reached the level of significance. These data suggest that there are subtle disruptions in cortical lamination

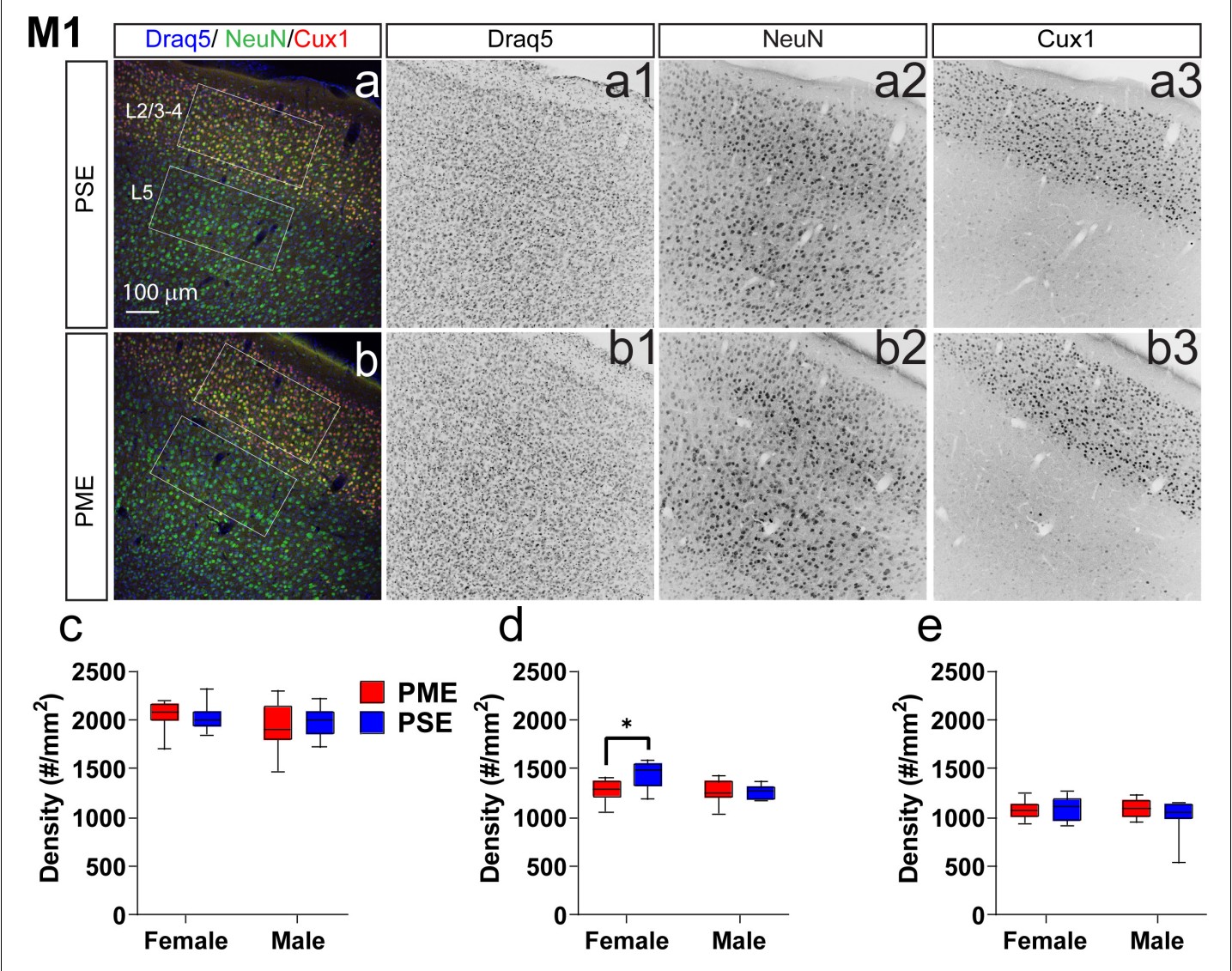

**Figure 7.** Neuronal density is reduced in the primary motor cortex (M1) of female PME offspring. Representative slices of the M1 in (a) PSE and (b) PME offspring demonstrating Draq5 (blue, (a1,b1): marker of nuclei), NeuN (green, (a2,b): marker of post-mitotic neurons), and Cux1 (red, (a3,b3): marker of upper cortical layers, typically layer 2/3–4). White boxes represent the areas used for quantification. (c) No effect of Cux1+ cell densities were observed in L2/3-4. (d) A significant decrease in NeuN+ cells was found in layer 2/3-4 of Female PME offspring (ANOVA: Interaction, p=0.044; Sidak's posthoc test, p=0.0098). (e) However, no differences were observed for NeuN+ cell densities in layer 5 (n = 9 (4M:5F) PME), 8 PSE (4M:4F). *p<0.05. Data were not collapsed on sex as analyses revealed both main-effects of sex and interactions with sex. Box plots indicate 25–75th percentiles with whisker characterizing the minimum and maximum value.

The online version of this article includes the following source data and figure supplement(s) for figure 7:

**Source data 1.** Numerical data to support graphs in *Figure 7* describing cortical lamination and neuronal density in M1.

**Figure supplement 1.** Development of the anterior cingulate cortex (ACC) is minimally affected by PME.

**Figure supplement 1—source data 1.** Numerical data to support graphs in *Figure 7—figure supplement 1* describing cortical lamination and neuronal density in ACC.

**Figure supplement 2.** Development of the primary somatosensory cortex (S1) is minimally affected by PME.

**Figure supplement 2—source data 1.** Numerical data to support graphs in *Figure 7—figure supplement 2* describing cortical lamination and neuronal density in S1.

development in PME for ACC, S1, and M1 regions with a notable reduction in neuronal density of the upper cortical layer of M1 in female PME.

## M1 neuronal intrinsic properties and local circuitry

Given the significant alterations in sensorimotor and locomotor behavior in offspring with PME, we next examined intrinsic properties, local circuitry, and morphology of layer 5B thick-tufted pyramidal neurons of the M1 in P21-P26 offspring using whole-cell patch clamp electrophysiology and laser scanning photostimulation for optical mapping of local circuitry (see *Figure 8a,b* for schematic diagram). Thick-tufted layer 5B neurons indicate neurons with specific subcortical projection targets (*Hattox and Nelson, 2007*; *Morishima et al., 2011*; *Oswald et al., 2013*), and most likely represented M1 pyramidal tract corticospinal neurons (*Suter et al., 2013*). Representative current-clamp traces from whole-cell recordings of action potential firing and sub-threshold voltage responses are shown in *Figure 8—figure supplement 1a* and *Figure 8c*, respectively. Recording analyses showed that L5 M1 neurons from PME mice displayed significantly reduced firing rates (number of APs) in response to injected current compared to PSE offspring (rmANOVA: Exposure, $F_{(1, 66)}=0.707$, p=0.40; Current, $F_{(2.198, 145.1)}=1109$ p<0.0001; Interaction, $F_{(12, 792)}=1.78$, p=0.047, *Figure 8—figure supplement 1b*). These findings suggest PME neurons are characterized by reduced intrinsic excitability at higher injected currents; however, no posthoc tests reached the level of significance. PME neurons exhibited significantly reduced input resistance compared to PSE cells (*Figure 8d*, see *Supplementary file 3* for stats), indicating that current was translated into a smaller change in voltage across the membrane of PME neurons. Additionally, subthreshold responses of L5 M1 neurons showed prominent 'sag' and 'overshoot' of the membrane potential (*Figure 8c*), which is characteristic of hyperpolarization-activated cyclic nucleotide-gated (HCN) channel current ($I_h$) expression (*Sheets et al., 2011*). Both voltage sag and voltage overshoot percentage were larger in L5 M1 neurons from PME mice (*Figure 8e,f*, see *Supplementary file 3* for stats), although voltage overshoot did not quite reach the level of significance. Additional intrinsic properties of L5 M1 neurons in offspring can be viewed in *Supplementary file 3*.

In addition to electrophysiological recordings, we measured the strength and organization of local synaptic inputs onto a single post-synaptic layer 5B M1 neuron by directing a UV laser beam (355 nm) to focally uncage glutamate across a 16 × 16 stimulation grid centered around the recorded neuron (*Figure 8b*). Compared to PSE (*Figure 8g,h*), PME (*Figure 8k,l*) modified the amplitude of local inputs onto layer 5 M1 neurons (see PME minus PSE amplitude difference images for females and males in *Figure 8i & m*, respectively). Row averages, which approximately correspond to subdivisions of cortex layers, were calculated by using the average amplitude of local inputs from rows 1:16, columns 6:13 of the stimulation grid. A significant interaction between exposure and row average was observed (rmANOVA: Exposure, $F_{(1,38)}=0.120$, p=0.73; Row, $F_{(3.759,142.8)} = 16.79$, p<0.0001; Sex, $F_{(1,38)}=5.168$, p=0.029; Row x Exposure, $F_{(15,570)}=2.411$, p=0.0021; Row x Sex, $F_{(15,570)}=0.561$, p=0.905; Exposure x Sex, $F_{(1,38)}=0.397$, p=0.532; Row x Exposure x Sex, $F_{(15,570)}=1.126$, p=0.329; *Figure 8j,n*). In particular, increased amplitude of excitatory responses were observed following stimulation of local neurons in L2/3 (*Figure 8i,j,m,n*). These electro-anatomical results suggest that the L2/3 → L5 local pathway is slightly enhanced in the M1 of juvenile PME mice.

Representative morphological reconstruction of recorded neurons can be viewed in *Figure 9a,b* (PSE and PME females, respectively) and *Figure 9d,e* (PSE and PME males, respectively) with all reconstructed thick-tufted L5 found in *Figure 9—figure supplement 1*. Sholl analyses suggest that morphology of layer 5 M1 neurons in PME mice is unchanged from PSE littermates as neither intersections (rmANOVA: Exposure, $F_{(1,27)}=0.602$, p=0.44; Radius, $F_{(16,432)}=148$, p<0.0001; Sex, $F_{(1,27)}=7.34$, p=0.012; Radius x Exposure, $F_{(16,432)}=0.632$, p=0.86; Radius x Sex, $F_{(16,432)}=3.87$, p<0.0001; Exposure x Sex, $F_{(1,27)}=0.0821$, p=0.78; Radius x Exposure x Sex, $F_{(16,432)}=0.666$, p=0.83; *Figure 9c,f top*, for males and females, respectively) nor length (rmANOVA: Exposure, $F_{(1,27)}=1.09$, p=0.30; Radius, $F_{(5.43,146.6)} = 119.6$, p<0.0001; Sex, $F_{(1,27)}=6.56$, p=0.016; Radius x Exposure, $F_{(16,432)}=0.659$, p=0.83; Radius x Sex, $F_{(16,432)}=3.10$, p<0.0001; Exposure x Sex, $F_{(1,27)}=0.305$, p=0.59; Radius x Exposure x Sex, $F_{(16,432)}=0.586$, p=0.89; *Figure 9c,f bottom*, for males and females, respectively) were different between PME and PSE treatments.

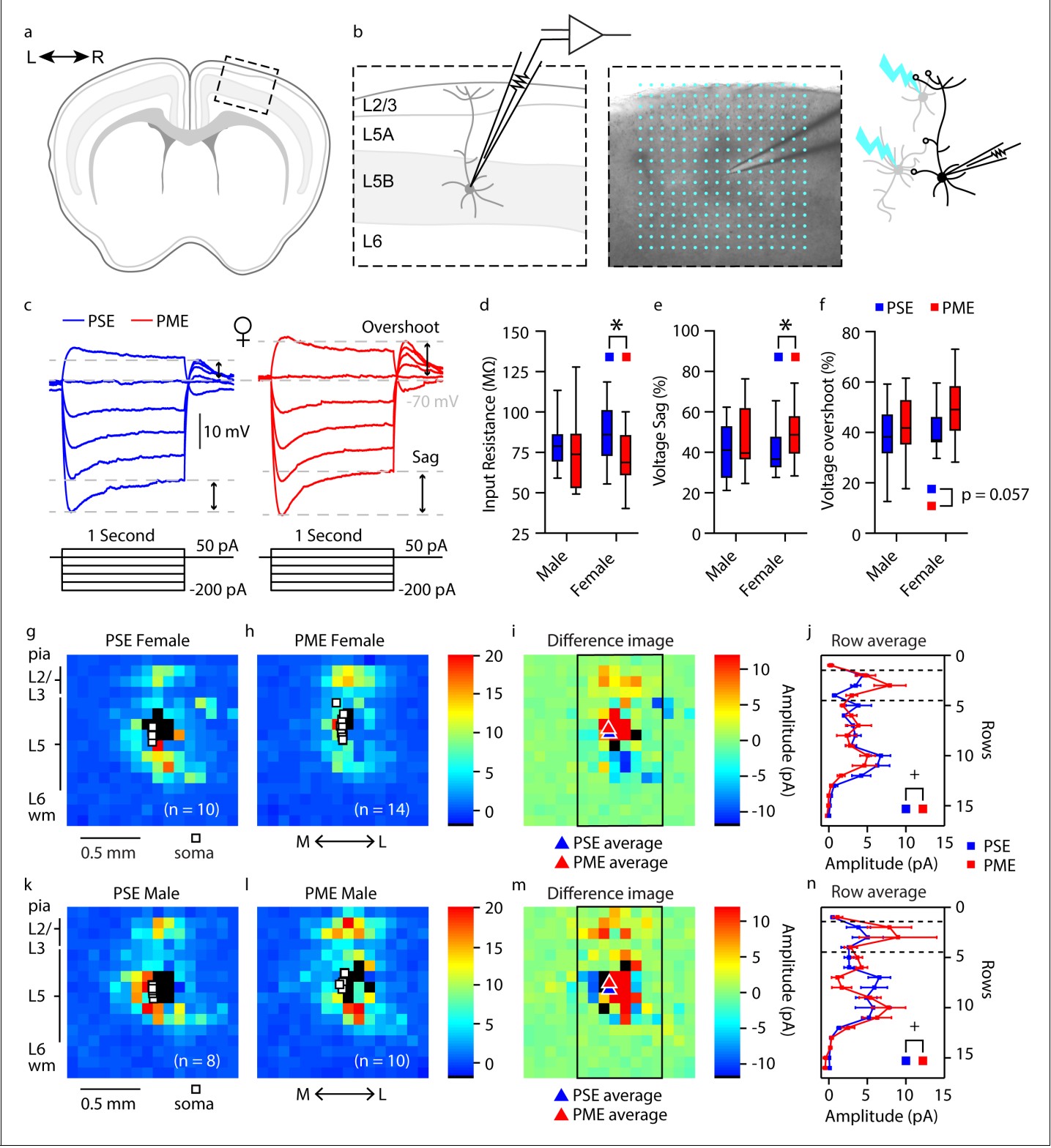

**Figure 8.** Intrinsic properties and local circuit mapping is altered by PME. (**a**) Brains were coronally sectioned to acquire M1 slices for patch clamp electrophysiology. (**b**) Representation of a M1 neuron recording (left) and example (4X bright field image) of a M1 neuron recording overlaid with the 16 × 16 stimulation grid (100 μm spacing) for local circuit mapping (center). Schematic of laser scanning photo-stimulation of local neurons connected to recorded L5 pyramidal neuron (right). (**c**) Representative current-clamp traces of sub-threshold responses from L5 M1 neurons from female mice. Step protocols and current values displayed below traces. (**d-f**) PME decreased input resistance (**d**), increased voltage sag (**e**), and increased voltage

*Figure 8 continued on next page*

Figure 8 continued

overshoots (**f**) (ANOVA: Exposure, p=0.030, p=0.027, and p=0.057; n = 13 PME mice (7M:6F), 41 cells (15M:26F) and n = 11 PSE mice (6M:5F), 28 cells (15M:13F)). (**g-h, k-l**) Average local input maps from PSE (n = 10 cells, 5 mice) and PME (n = 14 cells, 4 mice) female (**g,h**) and PSE (n = 8 cells, 5 mice) and PME (n = 10 cells, 5 mice) male (**k,l**) mice. Each pixel represents the mean amplitude of the response evoked by UV photolysis of MNI-glutamate at that location. (**i,m**) PME mice demonstrate differences in local circuitry as identified by the difference in PSE average map and PME average map for female (**i**) and male (**m**) mice. (**j,n**) This difference in local circuitry was further supported by a significant Row X Exposure (rmANOVA: Interaction, p=0.0025) indicating PME altered local synaptic input on L5 M1 neurons in a region dependent manner (n = 9 PME mice (5M:4F), 24 cells (10M:14F) and n = 10 PSE mice (5M:5F), 18 cells (8M:10F)). *p<0.05 for main effect of exposure. + p<0.05 for row by exposure interaction. Data were not collapsed on sex as analyses revealed both main-effects of sex and interactions with sex. Box plots indicate 25th to 75th percentiles with whisker characterizing the minimum and maximum value. Row average data points indicate mean ± SEM where rows are the average input from rows 1:16, columns 6:13.

The online version of this article includes the following source data and figure supplement(s) for figure 8:

**Source data 1.** Numerical data to support graphs in *Figure 8d–f* describing excitability and intrinsic properties.
**Source data 2.** Numerical data to support graphs in *Figure 8g–n* describing local circuit mapping in L5 M1 neurons.
**Figure supplement 1.** Current threshold for action potential (AP) firing in L5 M1 neurons is impacted by PME.
**Figure supplement 1—source data 1.** Numerical data to support graphs in *Figure 8—figure supplement 1* describing excitability of M1 L5 neurons.

## Discussion

Given the rapid rise in infants born passively exposed to opioids during prenatal development, there is an urgent need to develop translationally relevant animal models of prenatal opioid exposure to begin to elucidate the impact of opioid exposure on development. The present study adds to the

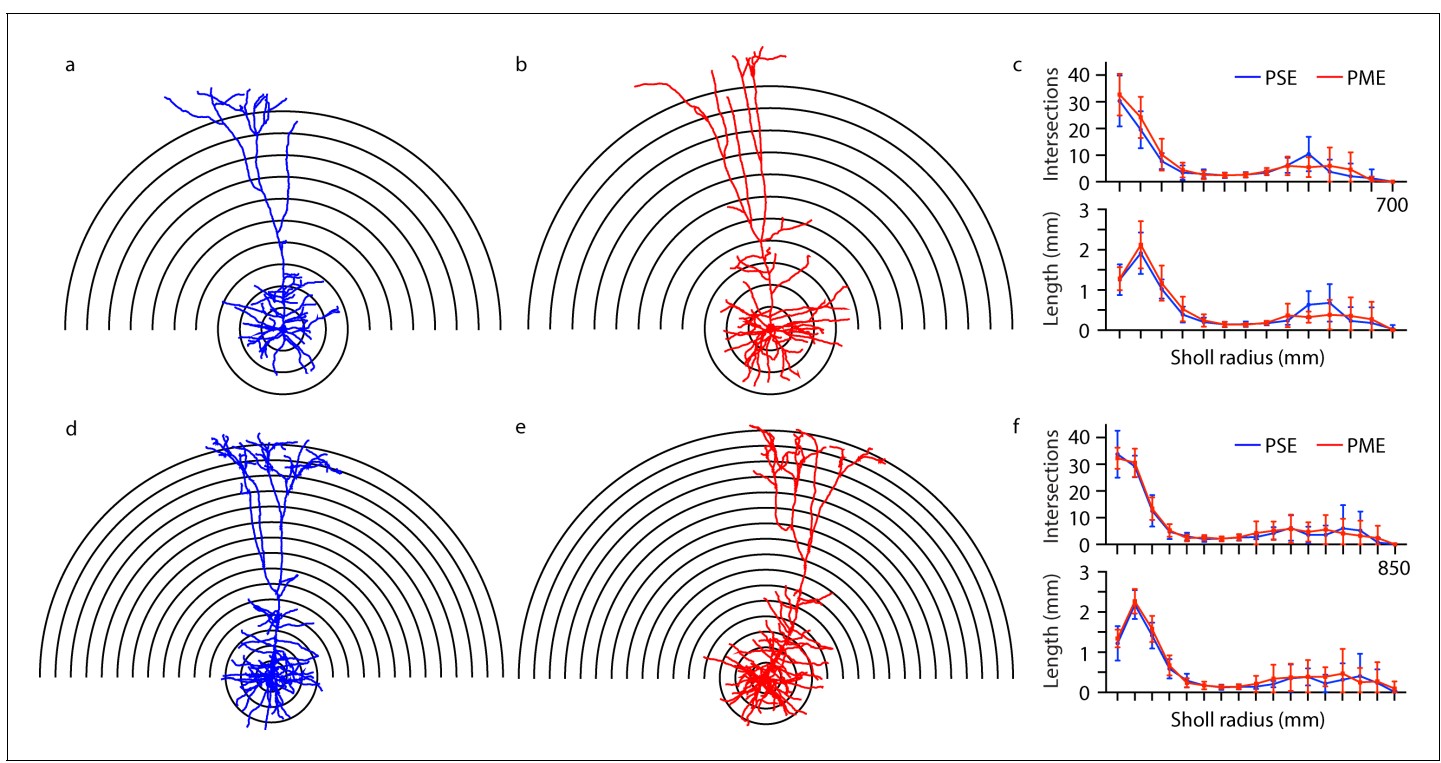

**Figure 9.** Morphological analysis of L5 M1 neurons revealed no effect of PME. (**a-b, d-e**) Representative morphological reconstruction and Sholl radii of motor cortex neurons filled with biocytin during whole-cell patch-clamp recordings in acute brain slices from PSE and PME female (**a–b**) and male (**d–e**) mice (concentric circles increase by 50 μm from 50 μm). (**c**) Analysis of PSE (n = 9) and PME (n = 10) intersections (top) and length (mm, bottom) in female mice by Sholl radius (μm, 50:50:700) revealed no significant differences related to exposure. (**f**) Analysis of PSE (n = 5) and PME (n = 7) intersections (top) and length (mm, bottom) in male mice by Sholl radius (μm, 50:50:850) also revealed no significant differences related to exposure. Data were not collapsed on sex as analyses revealed both main-effects or interaction with sex. Data points indicate mean ± SEM.

The online version of this article includes the following source data and figure supplement(s) for figure 9:

**Source data 1.** Numerical data to support graphs in *Figure 9* and *Figure 9—figure supplement 1* describing morphology of L5 M1 Neurons.
**Figure supplement 1.** Reconstruction of M1 L5 neurons in offspring.

limited body of work examining preclinical models of prenatal opioid exposure. We provide the first longitudinal assessment of behavioral and neuronal development in offspring with PME combined with data on maternal and fetal opioid levels in the brain, placenta, and blood. Furthermore, the present study builds upon previous models by providing a more comprehensive, clinically relevant characterization of how methadone, a commonly used opioid during pregnancy, impacts development when offspring are opioid-exposed during the entire gestational period. Our findings reveal a disruption in physical, behavioral, and neuronal development in PME male and female offspring which persists beyond the immediate neonatal period and supports clinical studies that suggest babies born with opioid exposure are at higher risk for adverse developmental outcomes (*Larson et al., 2019*; *Lee et al., 2020*; *Yeoh et al., 2019*).

Data on opioid levels in offspring are lacking in the preclinical literature. Bearing in mind differences in metabolism between rodents and humans, the methadone dose utilized here was likely in the clinical range (*Devidze et al., 2008*; *Dryden et al., 2009*). Although dam plasma levels were lower that what is typically recommended for treating OUD in humans (~400 ng/mL) (*Eap et al., 2002*), our dosing protocol produced opioid dependency in dams. Others also reported an accumulation of methadone in the rodent fetal brain (*Kongstorp et al., 2019*). We added to these findings by demonstrating methadone accumulated in the placenta and provided evidence that the placenta may have predictive power to infer the degree of prenatal brain opioid exposure during gestation. Although both methadone and EDDP accumulated in the placenta, EDDP did not concentrate in the fetal brain as methadone did. This observation may suggest the increase in methadone resulted from reduced metabolism in offspring. Methadone undergoes N-demethylation to form EDDP primarily via the cytochrome P450 enzyme CYP3A4 in humans (*Eap et al., 2002*), but the expression of most cytochrome P450 enzymes in both mice and humans are dramatically lower during the prenatal period (*Hart et al., 2009*). Nevertheless, we are confident that methadone was present in the fetal brain at pharmacologically relevant levels in our model and likely contributed to the aberrant neurobehavioral development.

The persistent decrease in the size of offspring was striking, and other models of prenatal opioid exposure, including methadone, similarly demonstrated reduced body weight (*Byrnes and Vassoler, 2018*; *Jantzie et al., 2020*; *Kunko et al., 1996*). In humans, prenatal opioid exposure is similarly associated with low birth weight and being small for gestational age (*Nørgaard et al., 2015*), but it remains unclear what underlies this reduced size. Given the delayed acquisition of other sensorimotor milestones, it is possible PME also delayed acquisition of the suckling reflex which could contribute to reduced growth. An alternative hypothesis is that methadone disrupted the hypothalamic–pituitary–somatotropic axis. Central administration of exogenous opioids modulates plasma levels of growth hormone and insulin-like growth factor in adult humans and rodents (*Abs et al., 2000*; *Hashiguchi et al., 1996*). Work done by Vathy and colleagues revealed that prenatal morphine exposure interacts with other hypothalamic-pituitary axes altering the homeostatic balance of stress-related hormones (*Rimanóczy et al., 2003*; *Slamberová et al., 2004*; *Vathy, 2002*), but the effect of prenatal opioid exposure on growth-related hormones has not been studied and may represent an important area of study for future work.

Separation-induced USVs emitted by mouse pups are frequently used as a measure of distress (*Ehret, 2005*). Production of these USVs is dependent on the endogenous opioid system. Mu opioid receptor (MOR) agonists reduce USV rates, and mice lacking MORs do not vocalize when separated from the dam (*Carden et al., 1991*; *Moles et al., 2004*). Others showed that PME reduces MOR-binding affinity and alters downstream signaling (*Kongstorp et al., 2020a*) suggesting the differences in USV development in our model resulted from changes in the endogenous opioid system. While the exact biological relevance of the increase in USV call rates and the alterations in call types and syntax on P7 is unclear at this time, these complex differences in USVs certainly reflected differential pup-dam communication between PME and PSE offspring.

PME offspring were hyperactive in the open field on P7 and P21 including a high number of jumps on P21. Combined with the increased USV call rate on P7, these activity findings are indicative of a heightened sensory arousal and hyperactive state in juvenile PME offspring. This hyperarousal phenotype may represent an enduring consequence of PME during the development of neurocircuitry controlling stress and/or sensorimotor activity. Our data support clinical findings that report higher risks of attention-deficit/hyperactivity disorder and increased hyperactivity and impulsivity in children with opioid exposure in utero (*Azuine et al., 2019*; *Ornoy et al., 2001*; *Sundelin Wahlsten*

*and Sarman, 2013*). In our model, PME offspring experienced delayed development of several sensorimotor milestones. Again, these findings mirror recent meta-analyses which support the negative impact of prenatal opioid exposure on psychomotor performance and motor function (*Lee et al., 2020*; *Yeoh et al., 2019*). In regards to animal models, others also reported delayed development of certain sensorimotor milestones (*Kunko et al., 1996*; *Robinson et al., 2020*; *Wallin et al., 2019*), but these reports slightly differ in which specific milestones are delayed and which develop normally in offspring. For instance, offspring from our PME model did not exhibit differences in negative geotaxis, although in other models of prenatal buprenorphine and methadone exposure, negative geotaxis was disrupted (*Kunko et al., 1996*; *Wallin et al., 2019*). Due to the differences in opioid used, duration and timing of exposure, and differences between rat and mouse development, identifying why specific milestones are delayed and why others are not remains challenging.

The neural mechanisms that could mediate the sensorimotor deficits and hyperactivity observed in our model are diverse. M1 has a central role in the preparation, execution, and adaptation of motor movements. Although the intrinsic properties of neurons can be affected by numerous ion channels and ion pumps, slice recording data indicated that altered intrinsic properties of M1 L5 pyramidal neurons from PME mice may be mediated by enhanced HCN channel-driven $I_h$. In cortical pyramidal neurons, there is a gradient of HCN channel expression with the greatest being in the apical dendrites (*Berger et al., 2001*; *Magee, 1998*; *Stuart and Spruston, 1998*). In mouse M1, HCN channels filter synaptic inputs onto L5 corticospinal neurons (*Anderson et al., 2010*; *Sheets et al., 2011*). As a result, increased $I_h$ in M1 L5 pyramidal neurons in PME mice suggests a narrowing of both spatial and temporal integration windows for both local and long-range synaptic inputs. Local circuit maps collected here showed that PME enhanced the L2/3→ L5 excitatory pathway in M1, which is the major local excitatory pathway for M1 (*Anderson et al., 2010*; *Weiler et al., 2008*). Therefore, it is possible that the narrowing of synaptic integration windows leads to compensatory mechanisms that enhanced the strength of synaptic inputs. These findings may suggest HCN channels represent a therapeutic target to prevent the potential neurobehavioral consequences of PME. In addition, methadone inhibits NMDA glutamate receptors (*Eap et al., 2002*), which mediate experience-dependent synaptic pruning during maturation of the mouse cortex (*Yu et al., 2013*). Therefore, PME-induced changes to dendritic pruning may also contribute to the altered local circuits of layer 5 M1 neurons, which could be further mediated by changes to HCN expression or function. While our morphological analysis revealed no effect of PME on L5 M1 neurons, measurements were highly variable indicating that neurons analyzed were a heterogeneous population of thick-tufted neurons.

Although methadone may be a necessary and effective treatment for maternal OUD, these findings suggest this treatment during pregnancy may come at the expense of persistent disruptions in offspring development. As clinical evidence strongly indicates the use of opioid maintenance therapies for treating OUD improves pregnancy outcomes (*ACOG, 2017*), these findings which indicate PME impairs physical and neurobehavioral outcomes in offspring should not be interpreted as evidence against the use of opioid maintenance therapies in pregnancy. Instead, early and preventative clinical intervention programs such as neurodevelopmental follow-up programs may be needed to support the healthy development of infants exposed to opioids in utero. Further research is necessary to determine if additional prenatal pharmacotherapies may prevent pathology and behavioral impairments in offspring. There are several limitations which must be considered when interpreting our findings. Prior reports indicated opioid treatment alters maternal behavior which could impact offspring development (*Slamberová et al., 2001*; *Wallin et al., 2019*). However, we and others (*Alipio et al., 2021*; *Kongstorp et al., 2020b*; *Tan et al., 2015*), did not observe differences in maternal care suggesting the differences found in offspring are a result of passive methadone exposure during gestational development. As maternal stress can have wide-ranging and persistent effects on physical, behavioral, and neurological development, additional longitudinal assessments of maternal care including maternal-offspring interactions throughout the preweaning period may provide additional information on maternal care. Alternatively, future work is necessary to examine how repeated opioid treatment may disrupt neuroendocrine physiology (e.g. HPA axis dysfunction or corticosterone production) which may indirectly disrupt offspring development. We cannot be confident that subtle differences in maternal behavior interacted with methadone exposure to alter offspring outcomes. We discovered few sex-specific effects in our assessments of offspring. This may be a result of the age at which the animals were tested or, if sex differences were very subtle, we

many have been inadequately powered to detect such small differences. Next, we modeled PME via noncontingent treatment using twice daily injections. Although free-access or self-administration would be more translational, noncontingent exposure reduces variability in methadone exposure between pregnant females. The interaction of PME with the stress of injections and repeated handling of dams could produce the differential offspring outcomes. Given that medically supervised opioid withdrawal is not recommended for OUD, our choice of prenatal opioid exposure model was based on epidemiological trends which supports the high usage of methadone by pregnant women (*Duffy et al., 2018*). However, there exists numerous variations in human opioid use during pregnancy which our model does not reflect (e.g. women may not initiate opioid maintenance therapy until partway through pregnancy; they may experience a relapse and/or use other recreational opioids at variable intervals during pregnancy; buprenorphine, which has a different pharmacological profile than methadone, is an increasingly frequent choice for opioid maintenance therapy). Further work by our group and others will be required to examine these variants of prenatal opioid exposure. The differences between rodent and human gestational development are an inherent limitation of our model that cannot be overcome. The first week of rodent postnatal life is roughly equivalent to the third trimester (*Robinson et al., 2020*). In an attempt to maintain a sufficient level of methadone exposure to the newborn pups during P0-P7, the dams were maintained at their highest gestational dose after giving birth and up to P7; however, it is unlikely this strategy completely replicates in utero third trimester opioid exposure in humans. Lastly, our findings in offspring were collected in the preweaning and early adolescent periods. Future studies will examine if alterations in physical, behavioral, and neuronal outcomes persist into adulthood or if PME offspring are able to eventually overcome these differences.

# Materials and methods

**Key resources table**

| Reagent type (species) or resource | Designation | Source or reference | Identifiers | Additional information |
|---|---|---|---|---|
| Strain, strain background (*Mus musculus* male and female) | C57BL/6J | Jackson Laboratories | B6/J | |
| Antibody | Rabbit anti-Cux1 (polyclonal) | Santa Cruz Biotechnology | Cat# SC-13024, RRID:AB_2261231 | (1:250 dilution) |
| Antibody | Mouse anti-NeuN (monoclonal) | Millipore | Cat# MAB377, RRID:AB_2298772 | (1:1000 dilution) |
| Antibody | Goat anti-mouse IgG (H+L), Alexa Fluor 488 conjugate | Thermo Fisher Scientific | Cat# A-11001, RRID:AB_2534069 | (1:1000 dilution) |
| Antibody | Goat anti-rabbit IgG (H+L), Alexa Fluor 555 conjugate | Thermo Fisher Scientific | Cat# A-32732; RRID:AB_2633281 | (1:1000 dilution) |
| Antibody | Streptavidin, Alexa Fluor 488 | Thermo Fisher Scientific | Cat# S32354; RRID:AB_2315383 | (1:1000 dilution) |
| Chemical compound, drug | Draq5 | Cell Signaling | Cat# 4084 | |
| Chemical compound, drug | Methadone HCl | National Institute on Drug Abuse | | |
| Chemical compound, drug | Oxycodone HCl | National Institute on Drug Abuse | | |
| Chemical compound, drug | MNI-caged glutamate | R and D Systems Inc (Tocris) | Cat# 295325-62-1 | |

*Continued on next page*

*Continued*

| Reagent type (species) or resource | Designation | Source or reference | Identifiers | Additional information |
|---|---|---|---|---|
| Chemical compound, drug | (R)-CPP | R and D Systems Inc (Tocris) | Cat# 126453-07-4 | |
| Software, algorithm | Prism 8.2 | GraphPad | | |
| Software, algorithm | Ethovision XT | Noldus | | |
| Software, algorithm | DeepSqueak | *Coffey et al., 2019* | | |
| Software, algorithm | Ephus software | Vidrio Technologies | | |
| Software, algorithm | ImageJ | NIH | | |
| Software, algorithm | Matlab | MathWorks | | |
| Software, algorithm | Neurolucida 360 | MBF Bioscience | | |
| Software, algorithm | Neurolucida Explorer | MBF Bioscience | | |
| Software, algorithm | PMOD Ma-Benveniste-Mirrione mouse brain atlas | PMOD Technologies | | |

## Animals and model generation

Animal care and research were conducted in accordance with guidelines established by the National Institutes of Health and protocols were approved by the Indiana University School of Medicine Institutional Animal Care and Use Committee. Eight-week-old female C57BL/6J mice were acquired from Jackson Laboratories (Bar Harbor, Maine), single housed, and randomly assigned to either saline (10 mL/kg) or oxycodone treatments. Oxycodone dependence was induced by a dose-ramping procedure with a dose of 10 mg/kg administered on pregestational day (PG) 14, 20 mg/kg on PG13, and then maintained on 30 mg/kg for PG12-6. All saline or oxycodone doses were administered subcutaneously twice daily at least 7 hr apart. On PG5, oxycodone-treated mice were transitioned to 10 mg/kg methadone while saline-treated animals continued to receive saline injections (s.c. b.i.d.). All oxycodone and methadone solutions were prepared in saline. Oxycodone and methadone were obtained from the National Institute on Drug Abuse Drug Supply Program. Five days following initiation of methadone treatment, an 8-week-old C57BL/6J male mouse (also acquired from Jackson Laboratories) was placed into the cage of each female for four days. Mucous plugs were assessed each morning to approximate gestational day (G) 0. Food consumption was examined weekly in a subset of these female mice. Female mice were weighed every Monday, Wednesday, and Friday throughout the study with the exception of the mating period.

Cages were examined for the presence of pups at the time of each morning and afternoon injection, and the day of birth was designated postnatal day (P) 0. Data on litter size, sexes, and neonatal deaths were collected. The presence of unconsumed placentas at 24 hr following birth was also examined. Only litters between three and eight pups were used in subsequent studies of offspring. In an attempt to maintain a sufficient level of methadone exposure to the newborn pups during P0-P7, the dams were maintained at their highest gestational dose after giving birth and up to P7. After P7, the dose of methadone administered to dams was adjusted to their body weight. All treatments to dams were discontinued at weaning (~P28).

## Methadone and metabolite concentrations

All tissue and blood samples were collected 2.5 hr following morning administration of methadone. On G18, three dams were anesthetized with ketamine/xylazine and fetuses were rapidly removed from the uterine cavity to examine brain and placental drug levels. On P1 and P7, three dams and their litters were sacrificed for the collection of trunk blood and brains. Blood samples were centrifuged for five minutes at 4500 g to collect 20 µL of plasma. Plasma, placenta, and whole brains were frozen in isopentane on dry ice and stored in −80°C until processing. The samples were examined for the presence of methadone and EDDP via high-performance liquid chromatography tandem mass spectroscopy (HPLC–MS/MS; Sciex 5500 QTRAP, Applied Biosystems, Foster City, MA). The analytical method for plasma samples has previously been described (*Metzger et al., 2020*). Tissue

samples and standards were prepared by adding diphenhydramine (internal standard, 0.1 ng/μL in methanol) and 0.1 M phosphate buffer (200 μL, pH 7.4) to tissue samples. Samples were then extracted using a liquid-liquid procedure with 3 mL ethyl acetate. The supernatant was transferred to a clean tube and evaporated to dryness, then reconstituted in acetonitrile with 0.1% formic acid (pH 6.5). Ten microliters of sample were then injected and the remaining protocol follows the previously published procedure (*Metzger et al., 2020*). The limit of quantification for methadone and EDDP detection was 0.1 ng/mL and 0.05 ng/mL in the plasma, respectively, and 0.08 ng/sample and 0.04 ng/sample of placenta and brain, respectively for methadone and EDDP.

## Maternal characteristics

Dams were given a new compressed paper nestlet, and twelve hours later, the ability of dams to build a new nest on P3 was assessed using a five-point nest rating system where one represents a nestlet untouched and a five requires >90% of nestlet to be torn creating a crater with walls higher than the height of the mouse for >50% of the circumference. Further criterion are described elsewhere (*Deacon, 2006*). A pup retrieval task was completed by first removing one pup and the dam and then placing the pup at one end of the cage (~30 cm away). The dam was then released at the other end of the cage and the time required to retrieve the missing pup was recorded. This was repeated with two other pups and an average pup retrieval latency score was calculated.

To demonstrate that the oxycodone dosing strategy induced dependency, a subset of mice underwent naloxone-precipitated withdrawal prior to methadone transition. Sixty to 90 min following the final 30 mg/kg oxycodone dose on PG6, all mice were administered naloxone (5 mg/kg, i.p.) and somatic signs of withdrawal were assessed for 10 min using a previously described protocol (*Berrendero et al., 2003*). These behavioral signs included number of paw shakes, wet dog shakes, and jumps and the presence of ptosis, body tremor, teeth chattering, piloerection, and diarrhea. The global withdrawal score was calculated using the following equation: (jumps*0.8+wetdogs shakes*1+diarrhea*1.5+paw shakes*0.35+ptosis*1.5+teeth chattering*1.5+body tremor*1.5+piloerrection*1.5). These mice which underwent precipitated withdrawal were not used in any subsequent studies. Similarly, following the weaning of offspring, a subset of dams underwent the naloxone-precipitated withdrawal procedure to demonstrate that the transition to the 10 mg/kg methadone dosing regimen maintained opioid dependency.

## Offspring physical development

Beginning at P0 and throughout the pre-weaning period, offspring were marked for identification on the paws with a red or black marker. The progression of weight gain and body length on P0, P3, P7, P10, P14, P18, and P21 was examined. A subset of the offspring was also weighed at P35 and P49. From P3 to P18, offspring were examined for the first day both eyes were open, the first day both ear pinnae completely unfolded and the ear canal was patent, and the first day the lower incisors emerged.

The right femur of P7 and P35 offspring was scanned on a SkyScan 1172 (Bruker, Billerica, MA, USA) with a 0.5 aluminum filter and a 6 μm voxel size. For P7 samples, mineralized femur length was obtained from a coronal view of the scan. Whole bone volume was obtained in the transaxial plane incorporating the entire mineralized bone. For P35 samples, a 0.5 mm region of interest at the distal femur was selected proximal to the growth plate. Distal femur metaphysis bone volume was obtained in the transaxial plane from this whole region including both trabecular and cortical bone. Trabecular bone was isolated in the same region for trabecular bone volume. Cortical bone in P35 samples was measured from five slices 2 mm proximal the end of the metaphysis region of interest. Cortical bone area and cortical thickness were measured at this site.

## Offspring behavior
### Open field

A modified open field (12.5 W x 15.8 L x 13.7 H cm) with a video recording device (Brio Ultra HD Pro Webcam, Logitech, Lausanne, Switzerland) and an ultrasound microphone (Pettersson Elektronik AB, Uppsala, Sweden) was used to assess locomotion and USVs development for five minutes on P1, P7, P14, and P21 in offspring. These sessions began at 8:00 A.M following a one-hour acclimation period. The arena was placed on a far infrared warming pad (Kent Scientific, Torrington, CT) that

maintained the surface temperature between 35 and 37°C. The entire apparatus was located within an environmental control chamber (Omnitech Electronics Inc, Columbus, OH) that was layered with sound proofing material. USV audio files were processed using the open-source, deep-learning-based system DeepSqueak (*Coffey et al., 2019*). Video tracking analysis was accomplished with EthoVision XT software (Noldus, Leesburg, VA). P1 recorded videos were examined for the number of twitches/jerks (defined as sudden and rapid muscle contractions of the limbs and/or whole body), a behavior consistent with NOWS in humans and rodents (*Kocherlakota, 2014*; *Ward et al., 2020*). As temperature instability is also a symptom of NOWS (*Kocherlakota, 2014*), body temperature was assessed on P1. Without a heating source, body temperature was monitored in offspring at baseline and after 2 min of isolation. Due to the small size of pups preventing collection of rectal temperatures, a FLIR E8 infrared camera (FLIR Systems, Wilsonville, OR) was used to capture surface body temperature. Post-hoc analysis of duration of time spent near the walls of the arena was assessed using Noldus software. Noldus Automatic Behavior Recognition was used to track the frequency of jumping, unsupported and supported rearing, and grooming behavior.

## Sensorimotor milestones

Beginning on P3 and through P14, offspring were tested daily to examine acquisition of: surface righting to prone position within 5 s of release, expression of negative geotaxis within 15 s of release demonstrated by ascending a slight incline after initial placement facing downwards, display of cliff aversion within 10 s of release on a ledge demonstrated by moving backward off the cliff edge, ability to grasp a suspended rod with both forelimbs for at least 2 s, and extinguishing of pivoting behavior demonstrated by walking outside of a 12 cm diameter circle within 30 s. These assessments are well-established paradigms used to characterize the normal development of newborn mice (*Hill et al., 2008*). These sessions began at 1:00 P.M. following a one-hour acclimation period. Acquisition of these sensorimotor developmental milestones was considered complete by a successful demonstration of the behavior within the allotted time for 3 consecutive days in a row.

## Acoustic startle and prepulse inhibition

Acoustic startle responses and PPI were assessed in P28-P29 mice in isolated chambers (SR-LAB Startle Response System, SD Instruments, San Diego, CA). Following five 115 Db startle pulses for acclimation, mice were exposed to eight blocks consisting of either a single pulse at the background level, 65, 75, 85, 95, 105, or 115 Db to assess acoustic startle response or a prepulse of either 65, 75, or 85 Db followed by a 115 Db startle pulse to assess PPI in random order. The intertrial interval was randomly assigned with intervals between 10 and 40 s. An accelerometer detected changes in force due to jumping/flinching and the output was an excitation voltage change in millivolts (mV).

## Magnetic resonance imaging

Volumetric MRI studies were performed using a 9.4 T Bruker system (Bruker BioSpin MRI GmbH, Germany) equipped with a BGA-12S gradient. A Bruker two-channel surface mouse head coil was used for high resolution brain imaging. Animals were anesthetized and maintained with 1.5–2% isoflurane during MRI sessions. T2-weighted images with a three-dimensional Rapid Acquisition with Relaxation Enhancement (RARE) sequence were acquired for volumetric measurements (TR/TE = 1500/11 ms, Rare Factor = 12, FOV = 20×20 x 7 mm with an isotropic voxel of 120 $\mu m^3$, number of averages = 1, scan time = 11.8 min).

A mouse brain atlas for 4-week-old mice was created by modifying PMOD (PMOD Technologies LLC, Switzerland) Ma-Benveniste-Mirrione mouse brain atlas. Specifically, the T2-weighted MRI images of a 4-week-old control mouse were normalized to the PMOD built-in mouse brain template using a Deform function in the 'Fuse It' module. VOIs were then manually adjusted to fit the 4-week-old mouse brain and to create a 4-week mouse atlas for this study. To obtain the brain volume measurements, T2-weighted anatomical images from individual mouse in the native space were normalized to the 4-week mouse template and an inverse transformation matrix was saved in Fuse It module. The native space image data were loaded again in the View module then 4-week-mouse atlas with VOIs was loaded and inverse transformed to obtain the volume measurements (see *Figure 6—figure supplement 1* for segmentation). Raw volume measurements were then utilized for statistical comparisons.

## Immunostaining

On P22-24, offspring were anesthetized with isoflurane and perfused with 4% paraformaldehyde prepared in PBS for 10 min at a pump rate of ~2 mL/min. Fixed brains were sectioned into 100 μm sections in the coronal plane using a Leica VT-1000 vibrating microtome (Leica Microsystems) and stored in PBS until later analysis. Sections were permeabilized with 0.3% Triton X100, then incubated with a blocking solution (3% normal goat serum prepared in PBS with 0.3% Triton X-100) and then incubated overnight with primary antibody (see *Key Resources Table*) prepared in blocking solution. An appropriate secondary antibody conjugated with an Alexa series fluorophore (see *Key Resources Table*) was used to detect the primary antibody. Draq5 (1:10,000 dilution, Cell Signaling) was included in the secondary antibody solution to stain nuclei. Z-stack confocal images were acquired from both hemispheres with a Leica confocal microscope with a 10X/NA0.75 objective. The Z-stacks were taken at 0.5 μm intervals, 5 μm-total thickness was imaged. Projection images of 5 μm thickness were used for image quantification by using NIH ImageJ software. The regions of interest were defined as bins 1–10 for the ACC and S1 and defined as layer 2/3–4 and layer 5 in M1.

## Electrophysiology, mapping, morphology, imaging

### Electrophysiology and mapping

At P21-26, acute brain slices were prepared as described previously (*Jones and Sheets, 2020*; *Sheets et al., 2011*). For acquiring the electrophysiological profiles of neurons, patch pipettes contained potassium-based intracellular solution (in mM: 128 K-gluconate, 10 HEPES, 1 EGTA, 4 MgCl$_2$, 4 ATP, and 0.4 GTP, 10 phosphocreatine, three ascorbate; pH 7.2). 3–4 mg biocytin (Sigma-Aldrich) was included in the intrapipette solution for morphological reconstruction. MNI-caged glutamate (Tocris Bioscience, 295325-62-1) [0.2 mM] was combined with recirculating solution for all recordings. The bath solution for recordings contained elevated concentrations of divalent cations [4 mM Ca$^{2+}$ and 4 mM Mg$^{2+}$] and NMDA receptor antagonist 3-(2-Carboxypiperazin-4-yl)propyl-1-phosphonic acid (CPP) [5 mM] (Tocris biosciences, 126453-07-4). Slices were used 1.5–4.5 hr after preparation. Recordings were performed at room temperature. Pipette series resistance was between 2 and 5 MΩ. Pipette capacitance was compensated; series resistance was monitored but not compensated and required to be <25 MΩ for inclusion in the data set. Current-clamp recordings were bridge-balanced. Current was injected as needed to maintain the membrane potential near −70 mV during select stimulus protocols. Recordings were filtered at 4 kHz and sampled at 10 kHz using a Multiclamp 700B amplifier (Molecular Devices). Membrane potential values were not corrected for a calculated liquid junction potential of 11 mV. Ephys software (http://www.ephus.org) was used for hardware control and data collection (*Suter et al., 2010*). Methods for determining input resistance, voltage threshold for action potential (AP) firing, and voltage sag have been reported previously (*Suter et al., 2013*). Glutamate uncaging and laser scanning photo-stimulation (glu-LSPS) were performed as described previously (*Cheriyan and Sheets, 2018*). Once a patch recording of a labeled neuron was established, an image of the slice (4X objective) was acquired before mapping for precise registration of the mapping grid. The mapping grid (16 × 16; 100 μm spacing) was rotated with the top row of the grid flush with the exterior primary motor area layer one. The grid locations were sampled (every 0.4 s) with a UV stimulus 1.0 ms in duration and 20 mW at the specimen plane. Photo-stimulation sites resulting in activation of glutamate receptors on the dendrites of the recorded neuron were readily detected based on characteristically short onset latencies (7 ms) of responses (*Anderson et al., 2010*) and included in the map analyses as synaptic responses resulting from uncaged glutamate activation of presynaptic neurons within the local circuit. Responses overlapped by direct activation of the recorded neuron's dendrites were excluded and rendered as black pixels in input color maps. These maps thus represent 'images' of the local sources of monosynaptic input arising from small clusters of ~100 neurons at each stimulus location. Excitatory (glutamatergic) responses were recorded at a command voltage of −70 mV. Excitatory input maps were constructed on the basis of the mean inward current over a 0–50 ms post-stimulus time window.

### Morphology

Following electrophysiology recordings, the pipette was removed slowly at an angle in voltage clamp mode while monitoring the capacitive transients in order to reestablish a seal. Following resealing of the final cell in a slice, the slice was removed from the recording chamber and left in

oxygenated ACSF solution for up to 3 hr, to ensure transport of biocytin to distal processes. The slice was then placed in 4% paraformaldehyde solution in 0.1 M Phosphate Buffer (4% PFA/PB) for 24 hr. Slices were washed (3x for 5 min at room temperature) with 0.1 M PBS and 0.1% Triton-X-100 (Acros Organics, AC21568-2500), then nutated in blocking solution containing normal goat serum (Millipore Sigma, 566380) for one hour. Slices were washed again, then nutated in blocking solution with secondary antibody Streptavidin Alexafluor conjugate 488 (1:1000; Fisher Scientific, S32354) at room temperature for 1 hr. Slices were washed and mounted to glass slides. Images of recovered biocytin-filled neurons were taken using a Nikon A1R confocal microscope with ×40 oil immersion objective. Z-stack images were taken at a pitch of 1.5 μm and stitched together with 15% overlap. Neurolucida and Neurolucida 360 (MBF Bioscience) was used for reconstruction and Sholl analysis.

### Data exclusion criteria
Of 125 neurons, 10 were excluded from analysis due to pipette series resistance >25 MΩ and instability over time, six of which were PME-treated neurons from male mice. After morphological reconstruction and image acquisition, additional inclusion criteria were established: (1) imaged M1 neurons with no pia-directed dendrite were excluded from all analyses (6/115), (2) imaged M1 neurons with no apical tuft at the end of a pia-directed dendrite were excluded from excitatory input map analysis on the assumption that vibratome slicing separated apical tuft from cell body (8/115) and (3) M1 neurons which did not reseal and were unable to be imaged (5/115), or neurons which did not have 'maps' collected (11/115), were excluded from excitatory input map analysis. 20/115 neurons were not processed for morphological reconstruction but were included in electrophysiology intrinsic data analysis. Thin-tufted neurons (23) were excluded from analysis on the basis of <20% voltage sag. Forty-two-thick-tufted neurons were utilized for map analysis and of these, 31 were reconstructed for morphological analysis.

## Statistics
Experimenters were blinded to treatment/exposure group for data collection of all studies. Statistical analyses were conducted using GraphPad Prism 8 (San Diego, CA). Data are graphically presented as the mean ± SEM or as box plots extending from the 25th to 75th percentiles with whisker characterizing minimum and maximum values. The level of significance was set at $p < 0.05$. All experiments were performed using both male and female offspring. To minimize potential litter effects in all completed studies, offspring from four or more litters per exposure were utilized as previously done (*Ricalde and Hammer, 1990*). For offspring weight and length, offspring from eight to nine litters per exposure were examined. For all behavioral studies, offspring were taken from four to five different litters per exposure. For neuroimaging and immunohistochemistry studies, offspring from four litters per exposure were utilized. For electrophysiology, mapping, morphology, and imaging, neurons were recorded and imaged from offspring of nine different litters per exposure. With the exception of MRI data, all studies were sufficiently powered to detect sex differences. Therefore, all analyses (with the exception of MRI data) were first completed treating sex as a factor. However, for clarity of focusing on the effect of prenatal exposure, when no main effects of sex or interactions with sex were found in offspring, the data was collapsed on sex and re-analyzed. Chi-square tests for categorical data was used for litter characteristics. A linear regression was performed to assess the relationship between placental methadone levels and offspring brain levels on G18, and offspring plasma and brain levels on P1 and P7. Two-tailed unpaired t-tests were used to analyze normally distributed data and Mann-Whitney U tests or Wilcoxon rank sum test (for electrophysiology data only) were used to analyze non-normally distributed data. For data with multiple groups and/or repeated measures, ANOVAs with Sidak's post hoc tests were used.

## Acknowledgements
We thank the National Institute on Drug Abuse Drug Supply Program for generously providing the methadone and oxycodone utilized in the experiments of this manuscript. Mass spectrometry work was provided by the Clinical Pharmacology Analytical Core at Indiana University School of Medicine; a core facility supported by the IU Simon Cancer Center Support Grant P30 CA082709. We thank Dr. David McKinzie, the director of the Behavioral Phenotyping Core at Indiana University School of Medicine, for valuable comments on study design.

## Additional information

### Funding

| Funder | Grant reference number | Author |
|---|---|---|
| National Institute on Alcohol Abuse and Alcoholism | R01AA027214 | Brady K Atwood |
| National Institute on Alcohol Abuse and Alcoholism | F30AA028687 | Gregory G Grecco |
| National Institute on Alcohol Abuse and Alcoholism | T32AA07462 | David L Haggerty Kaitlin C Reeves |
| Indiana University | | Bryan K Yamamoto Hui-Chen Lu Brady K Atwood |
| Indiana University Health | | Brady K Atwood |
| IU Simon Cancer Center | | Andrea R Masters |
| Stark Neurosciences Research Institute | | Gregory G Grecco David L Haggerty Brady K Atwood |

The funders had no role in study design, data collection and interpretation, or the decision to submit the work for publication.

### Author contributions

Gregory G Grecco, Conceptualization, Resources, Data curation, Formal analysis, Funding acquisition, Investigation, Visualization, Methodology, Writing - original draft, Writing - review and editing; Briana E Mork, Hunter Hoffman, Andrea R Masters, Cameron W Morris, Data curation, Formal analysis, Writing - review and editing; Jui-Yen Huang, Corinne E Metzger, Data curation, Formal analysis, Investigation, Writing - review and editing; David L Haggerty, Formal analysis, Methodology, Writing - review and editing; Kaitlin C Reeves, Investigation, Writing - review and editing; Yong Gao, Simon N Katner, Erin A Newell, Jiuen Kim, Data curation, Writing - review and editing; Eric A Engleman, Supervision, Writing - review and editing; Anthony J Baucum, Bryan K Yamamoto, Patrick L Sheets, Conceptualization, Supervision, Writing - review and editing; Matthew R Allen, Conceptualization, Project administration, Writing - review and editing; Yu-Chien Wu, Hui-Chen Lu, Formal analysis, Supervision, Writing - review and editing; Brady K Atwood, Conceptualization, Supervision, Funding acquisition, Investigation, Methodology, Writing - original draft, Project administration, Writing - review and editing

### Author ORCIDs

Gregory G Grecco (ID) https://orcid.org/0000-0002-0700-8633
Briana E Mork (ID) http://orcid.org/0000-0002-5249-3738
Jui-Yen Huang (ID) http://orcid.org/0000-0003-4745-9970
David L Haggerty (ID) http://orcid.org/0000-0002-1455-2557
Hui-Chen Lu (ID) http://orcid.org/0000-0002-6628-7177
Brady K Atwood (ID) https://orcid.org/0000-0002-7441-2724

### Ethics

Animal experimentation: The animal experimental procedures in this study were approved by the Institutional Animal Care and Use Committee at the Indiana University School of Medicine (Protocol Number 19017). Guidelines set forth by the National Institutes of Health (Maryland, USA) for ethical treatment and care for experimental animals were followed. Whenever possible, we sought to minimize pain and distress of animals. Euthanasia was only performed on mice that were under a deep plane of anesthesia (achieved using isoflurane) which was assessed via the pedal withdrawal reflex.

Decision letter and Author response
Decision letter https://doi.org/10.7554/eLife.66230.sa1
Author response https://doi.org/10.7554/eLife.66230.sa2

## Additional files

**Supplementary files**

• Supplementary file 1. Dam and offspring methadone and metabolite concentrations. n = 3 dams + their respective litters per timepoint; n = 17–20 offspring samples at G18, n = 15 offspring at P1, n = 17–18 offspring at P7. All tissue and blood samples were collected 2.5 hr following the morning administration of methadone. Data are collapsed across offspring sex. All data are mean ± SEM. EDDP: 2-ethylidene-1,5-dimethyl-3,3-diphenylpyrrolidine. The limit of quantification for methadone and EDDP detection was 0.1 ng/mL and 0.05 ng/mL in the plasma, respectively, and 0.08 ng/sample and 0.04 ng/sample of placenta and brain for both methadone and EDDP.

• Supplementary file 2. Test statistics for volumetric MRI analysis. Unpaired t tests, n = 11 (4M:7F) PME, 11 PSE (6M:5F) mice. See *Figure 8—figure supplement 1* for visual representation of the data. *R,* Right; *L,* Left

• Supplementary file 3. Intrinsic properties of L5 M1 neurons. Resting membrane potential evaluated with no applied current, all other properties evaluated with current applied to hold the membrane potential near minus 70 mV. Data were not collapsed on sex as analyses revealed some main-effects of sex. Data presented as mean ± SEM. F statistics (df = 1,65) are presented in the final column with significant results bolded (*p<0.05). n = 10 PME mice (6M:4F), 30 cells (13M:17F) and n = 9 PSE mice (5M:4F), 23 cells (11M:12F).

• Transparent reporting form

### Data availability

All data generated or analyzed during this study are included in the manuscript and supporting files. Source data files have been provided for all figures and tables.

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
