## [Decision Letter]

Thank you for submitting your article "Prenatal Methadone Exposure Disrupts Behavioral Development and Alters Motor Neuron Intrinsic Properties and Circuitry" for consideration by *eLife*. Your article has been reviewed by three peer reviewers, and the evaluation has been overseen by Alicia Izquierdo as the Reviewing Editor and Michael Taffe as the Senior Editor. The following individuals involved in review of your submission have agreed to reveal their identity: Lauren L Jantzie (Reviewer #1); W Christopher Risher (Reviewer #2).

Essential Revisions:

Reviewers request edits that fall in the category of additional analyses, methodological details, and discussion. Reviewers do not request additional experiments.

1) There needs to be clarification of sex differences/analyses. The authors included both sexes in many of their analyses but it is not always clear when the sex of the offspring were combined in the analyses and/or whether sex was always included as a factor in the many endpoints.

2) The analysis of structural differences measured by volumetric MRI showed that there were no appreciable differences across grey matter structures with PME. However, these deficits tend to be more true for white matter than grey, though the authors do not indicate whether this was investigated. Please clarify with additional analyses.

3) Reviewers request added details related to drug preparation/administration, and the timeline of administration. Several of the methods descriptions referenced details published elsewhere, but brief descriptions should be included in the present manuscript. It is also unclear if regional measurements of brain regions were adjusted according to intracranial/total brain volume and body weight.

4) Though reviewers do not request additional data or experiments related to this, authors should note potential ways that maternal stress could weigh in to the present results and how they could be addressed in the future.

Reviewer #1 (Recommendations for the authors):

1) The authors should describe how the methadone was made and the vehicle it was made up in, including any diluents and excipients.

2) In the Discussion it would be helpful to have the authors describe how they see this data changing the field or the clinical implications of the findings with greater depth. What are some suggested solutions? Neurorepair? Neurodevelopmental follow-up? How should these infants be treated in the NICU? This is briefly touched on but incompletely described.

3) It would be beneficial to describe sex differences observed.

Reviewer #2 (Recommendations for the authors):

1) Several of the methods descriptions merely referenced details published elsewhere (e.g. assessments of nest building, somatic withdrawal assessment, hyperthermia, myoclonic jerks). Brief descriptions should be included here.

2) Were regional measurements of brain regions adjusted according to intracranial/total brain volume and body weight?

3) A previous study using C57Bl/6J mice showed that prolonged, pre-test handling and/or sham injections was sufficient to curtail the early stress response inherent to this strain (Ryabinin et al., 1999 [DOI 10.1016/S0091-3057(98)00239-1]. In the absence of such a control group for all included studies, however, I would at least like to have seen measurements of stress markers (e.g. c-fos, cortisol) in dams with the current dosing paradigm as well as age-matched controls that either did not receive any injections or that underwent some type of anti-stress precautions.

Reviewer #3 (Recommendations for the authors):

A lack of coordination during development would be best tested using a rotarod or related tasks.

For Figure 6, it is not clear whether the results were analyzed for males and females separately. Was the impact of PME on cell density driven by one or the other sex? This is relevant given the sex-specificity of some of the other phenotypes reported, including USVs.

Generally, in several of the figures/analyses it was unclear whether and how sex was included as a factor. Some examples that need clarification: Figure 7; Figure 8; Figure 5—figure supplement 2, Figure 5—figure supplement 5; Figure 6—figure supplement 1; Figure 6; Figure 7—figure supplement 1.

Based on the number of animals represented in each group for some of these experiments, it seems that the authors were not properly powered to examine potential sex differences. This is in contrast to some of the other findings reported. Making that clearer in the manuscript would strengthen the article. It should also be noted as a potential caveat in the Discussion.

---

## [Author Response]

Essential Revisions:Reviewers request edits that fall in the category of additional analyses, methodological details, and discussion. Reviewers do not request additional experiments.1) There needs to be clarification of sex differences/analyses. The authors included both sexes in many of their analyses but it is not always clear when the sex of the offspring were combined in the analyses and/or whether sex was always included as a factor in the many endpoints.

We apologize for this confusion and agree that the interpretation of the analyses could be improved with better clarification. All analyses, with the exception of the MRI analysis, were powered to detect sex-effects and the analyses were first completed including sex as a factor (Although, we were originally underpowered to detect sex-effects for immunostaining data, we have included additional replicates in this resubmission which has provided us with the power to detect sex-effects (Figure 7 and Figure 7—figure supplements 1 and 2). When sex-effects were discovered, they were stated in the Results section and data was represented separated by sexes in figures. When no main effects of sex or interactions with sex were found, the factor was collapsed and the data were re-analyzed, no longer including sex as a factor. Figures were then represented collapsed on sex. Given the large amount of data collected and analyzed, our goal was to ensure the reader was able to focus on prenatal exposure effects as opposed to bogging them down in many null effects of sex. This was originally stated in the “statistics” section of the Materials and methods but was further clarified in this resubmission. To further clarify our analysis approach, we briefly acknowledge within each figure legend how we handled sex in our analyses.

**2) The analysis of structural differences measured by volumetric MRI showed that there were no appreciable differences across grey matter structures with PME. However, these deficits tend to be more true for white matter than grey, though the authors do not indicate whether this was investigated. Please clarify with additional analyses.**

Although human studies have certainly demonstrated grey matter deficits (e.g. Hartwell et al., 2020, cited in the original submission as well as Yuan et al., 2014 (DOI: 10.1038/jp.2014.111); Walhovd et al., 2007 (DOI: 10.1016/j.neuroimage.2007.03.070); Sirnes et al. 2017, white matter deficits have also been discovered in both rodent models and in human children with prenatal opioid exposure. In contrast to gray matter, white matter tracts are more or less “continuous” making volumetric approaches to white matter changes less than ideal. A more appropriate approach to the examination of white matter structure would be DTI tractography which we unfortunately did not collect during this scan session. Completing DTI would almost certainly be too large of an undertaking for the present manuscript; nonetheless, we are actively interested in examining white matter deficits using DTI and hope to complete this method in future studies using our mouse model of PME.

3) Reviewers request added details related to drug preparation/administration, and the timeline of administration. Several of the methods descriptions referenced details published elsewhere, but brief descriptions should be included in the present manuscript. It is also unclear if regional measurements of brain regions were adjusted according to intracranial/total brain volume and body weight.

We apologize for the lack of critical details and have added additional detail on drug preparation, drug administration, and the timeline of model preparation including adding additional information to Figure 1A (and also Materials and methods). Furthermore, we have expanded our Materials and methods section in several areas with greater detail on experiments completed. We did *not* normalize grey matter volumes to intracranial/total brain volume and body weight. Given that PME animals were smaller at the time of scanning (~P28) and differences were not present using raw volumes, we feel that normalization could falsely inflate region grey matter differences in PME animals. We have now clarified that these values were not normalized to body weight or intracranial/total brain volume (Materials and methods).

**4) Though reviewers do not request additional data or experiments related to this, authors should note potential ways that maternal stress could weigh in to the present results and how they could be addressed in the future.**

We strongly agree with the reviewers that increased maternal stress could impact offspring neurodevelopmental outcomes and thank them for drawing particular attention to this potential limitation of our model. As we were developing this rodent model of PME, the authors shared this concern regarding the effects of repeated handling and injections on maternal stress levels which could, in turn, negatively affect offspring outcomes. Admittedly, we did not utilize age-matched completely naïve (e.g. no injections or handling) dams as a non-stressed control group for all studies would have required a substantial number of additional animals that may not have added significant value to the interpretation of the data. Our primary interest was to examine the effects of prenatal drug exposure, and, therefore, our greatest concern was to ensure that maternal behaviors did not differ between opioid-injected and saline-injected dams. As can be observed in Figure 3, methadone dams demonstrate remarkable resiliency to repeated methadone injections relative to the saline injected animals, although opioid-injected dams are physically dependent on opioids (Figure 3E). This certainly translates to the “bedside” as the purpose of methadone treatment is to reduce opioid craving and curtail withdrawal allowing individuals to live their everyday lives. Therefore, we feel it is unlikely that increased maternal stress among opioid-treated dams contributed to the outcomes we describe in PME offspring. Nonetheless, we previously acknowledged this limitation in our original submission and have expanded on this limitation as requested (Discussion).

Reviewer #1 (Recommendations for the authors):

1) The authors should describe how the methadone was made and the vehicle it was made up in, including any diluents and excipients.

We apologize for oversight and have included in this resubmission that the methadone solution was dissolved in saline (Materials and methods).

**2) In the Discussion it would be helpful to have the authors describe how they see this data changing the field or the clinical implications of the findings with greater depth. What are some suggested solutions? Neurorepair? Neurodevelopmental follow-up? How should these infants be treated in the NICU? This is briefly touched on but incompletely described.**

We thank the reviewer for the opportunity to further discuss the implications of our findings. Our original intention was not to overstate our findings and we remain cautious regarding suggested solutions for additional prenatal treatments as our findings do not directly point to a drug target at this point. Nonetheless, we have expanded our concluding paragraph to further discuss our findings in a clinical context (Discussion). <bold />*3) It would be beneficial to describe sex differences observed.*We apologize for the confusion regarding our analyses and hope we have made significant improvements in our description of analyses. When sex-effects were discovered they were discussed in detail; however, as described above, the sex-effect were relatively few, and when there were no sex-effects, sex was collapsed as a factor, and the data were re-analyzed (see reply to critique #1 above).

Reviewer #2 (Recommendations for the authors):

1) Several of the methods descriptions merely referenced details published elsewhere (e.g. assessments of nest building, somatic withdrawal assessment, hyperthermia, myoclonic jerks). Brief descriptions should be included here.

We apologize for the lack of detail and have included further description of these assessments in the following resubmission (see our response to overarching review point #3).

2) Were regional measurements of brain regions adjusted according to intracranial/total brain volume and body weight?

As described above, we chose not to normalize grey matter volumes to intracranial/total brain volume and body weight. Had we observed a reduction in grey matter ROIs as prior human studies have demonstrated, normalizing to body weight or total brain volume may have been appropriate. Given that no grey matter volume reductions were observed for the raw data, we feel that adjusting to body weight or total brain volume will not provide any added benefit when interpreting the results of the manuscript. We have acknowledged that we did not normalize these values in the resubmission (Materials and methods).

3) A previous study using C57Bl/6J mice showed that prolonged, pre-test handling and/or sham injections was sufficient to curtail the early stress response inherent to this strain (Ryabinin et al., 1999 [DOI 10.1016/S0091-3057(98)00239-1]. In the absence of such a control group for all included studies, however, I would at least like to have seen measurements of stress markers (e.g. c-fos, cortisol) in dams with the current dosing paradigm as well as age-matched controls that either did not receive any injections or that underwent some type of anti-stress precautions.

We strongly agree with the reviewers that increased maternal stress could impact offspring neurodevelopmental outcomes. Although, we did not observe differences in pregnancy rates, maternal weights, or maternal care behaviors we assessed, it remains plausible that the twice daily injections could impact maternal stress levels which could produce subtle effects on maternal behaviors which interacts with PME to alter offspring outcomes. We rightfully acknowledged this limitation in the original submission and have further expanded on this limitation to our work and included potential ways this limitation could be addressed by others in the future (Discussion).

Reviewer #3 (Recommendations for the authors):A lack of coordination during development would be best tested using a rotarod or related tasks.

We agree that tests of motor coordination are important to use for our model. We intend to use these for our future studies.

For Figure 6, it is not clear whether the results were analyzed for males and females separately. Was the impact of PME on cell density driven by one or the other sex? This is relevant given the sex-specificity of some of the other phenotypes reported, including USVs.Generally, in several of the figures/analyses it was unclear whether and how sex was included as a factor. Some examples that need clarification: Figure 7; Figure 8; Figure 5—figure supplement 2, Figure 5—figure supplement 5; Figure 6—figure supplement 1; Figure 6; Figure 7—figure supplement 1.

We apologize for this confusion on how our analyses were completed. As described in detail above, we have now provided a brief description to the figure legends on how sex was assessed within each figure legend.

**Based on the number of animals represented in each group for some of these experiments, it seems that the authors were not properly powered to examine potential sex differences. This is in contrast to some of the other findings reported. Making that clearer in the manuscript would strengthen the article. It should also be noted as a potential caveat in the Discussion.**

We thank the reviewer for noting this potential limitation to our work. Although every effort was made to include an equal number and appropriate numbers of sexes in our studies so that we were sufficiently powered to detect sex differences, in some studies, additional numbers of animals may have increased the power to detect potential sex differences in offspring outcomes. ≥7 mice per sex per exposure were utilized in behavioral experiments, ≥9 mice per sex per exposure were utilized for weight and length assessments, ≥5 mice per sex per exposure were utilized for bone density studies, ≥11 cells per sex per exposure were utilized in electrophysiological assessments, and ≥6 cells per sex per exposure were utilized in morphology studies. For this resubmission, we were able to include more animals for our immunostaining analysis of cell density in ACC, S1, and M1 which provided the power to detect sex-effects (≥4 mice per sex per exposure); however, we do not have the appropriate number of animals to examine sex effects for the grey matter volumes assessed via MRI which is why this data was not analyzed with sex as a factor. We have now provided a statement in the limitations paragraph of our Discussion indicating that increased power may have revealed additional sex-effects.